# Graded recruitment of pupil-linked neuromodulation by parametric stimulation of the vagus nerve

Zakir Mridha[1,2,6], Jan Willem de Gee [1,2,6], Yanchen Shi [1,2], Rayan Alkashgari[3], Justin Williams [3], Aaron Suminski[3], Matthew P. Ward[4], Wenhao Zhang[1,2] & Matthew James McGinley [1,2,5 ✉]

Vagus nerve stimulation (VNS) is thought to affect neural activity by recruiting brain-wide release of neuromodulators. VNS is used in treatment-resistant epilepsy, and is increasingly being explored for other disorders, such as depression, and as a cognitive enhancer. However, the promise of VNS is only partially fulfilled due to a lack of mechanistic understanding of the transfer function between stimulation parameters and neuromodulatory response, together with a lack of biosensors for assaying stimulation efficacy in real time. We here develop an approach to VNS in head-fixed mice on a treadmill and show that pupil dilation is a reliable and convenient biosensor for VNS-evoked cortical neuromodulation. In an 'optimal' zone of stimulation parameters, current leakage and off-target effects are minimized and the extent of pupil dilation tracks VNS-evoked basal-forebrain cholinergic axon activity in neocortex. Thus, pupil dilation is a sensitive readout of the moment-by-moment, titratable effects of VNS on brain state.

---

[1] Department of Neuroscience, Baylor College of Medicine, Houston, TX, USA. [2] Jan and Dan Duncan Neurological Research Institute, Texas Children's Hospital, Houston, TX, USA. [3] Department of Biomedical Engineering, Madison, WI, USA. [4] Department of Biomedical Engineering, Purdue University, West Lafayette, IN, USA. [5] Department of Electrical and Computer Engineering, Rice University, Houston, TX, USA. [6] These authors contributed equally: Zakir Mridha, Jan Willem de Gee. ✉email: matthew.mcginley@bcm.edu

The vagus nerve carries efferent parasympathetic and afferent sensory information between the brain and body. Early work showed that stimulation of the vagus nerve (VNS) caused gross changes in the electroencephalogram[1] and acutely suppresses seizure activity[2]. For over 30 years since then, VNS has been in wide use as an epilepsy treatment[3]. In addition, VNS is currently explored in other brain disorders, such as major depression, anxiety, autism, tinnitus, and Alzheimer's disease[4–13]. Despite its wide and growing use, the mechanisms by which VNS exerts its clinical benefits are still largely unknown.

Functional neuroimaging in humans and *c-fos* imaging in rats show VNS-evoked activity in many brain regions, including the amygdala, thalamus, hypothalamus, and cerebral cortex[14,15]. This widespread brain activation pattern, together with the functional anatomy of the direct connections of the vagus nerve with the brain, suggests a prominent role for brain-wide neuromodulatory systems in mediating the effects of VNS on the brain. Indeed, lesions of the noradrenergic locus coeruleus (LC) occlude the seizure-attenuating effects of VNS[16], and VNS induces phasic firing in LC neurons that increases in intensity with increasing VNS pulse width and amplitude[17,18]. Furthermore, the effects of VNS on cortical excitability synchrony are occluded by blockade of muscarinic receptors in the auditory cortex, suggesting a role for acetylcholine released by the basal forebrain[19]. However, the coarse occlusion in these lesion and pharmacological studies may result from unphysiological knock-on effects that do not reflect the natural mechanisms. Therefore, whether and which neuromodulators mediate the therapeutic benefits of VNS is largely unknown.

In sensory and motor systems, VNS-evoked cortical neuromodulation is increasingly explored for therapeutic benefits in a Hebbian framework, by temporally pairing VNS with sensory stimulation or movements. These studies suggest a prominent role for VNS-evoked basal forebrain acetylcholine release in the neocortex. On the motor side, VNS paired with forelimb movements drives reorganization of representations in the motor cortex, and the plasticity is lost after lesions of cortical cholinergic projections from the basal forebrain[20]. On the sensory side, pairing sound presentation with VNS results in auditory cortical map reorganization similar to that observed when sound is paired with nucleus basalis stimulation[21,22]. However, it has not been demonstrated whether VNS actually evokes acetylcholine release from the basal forebrain into the cortex, so the role of cortical acetylcholine in VNS-evoked cortical plasticity remains speculative.

A major impediment to parsing mechanisms, optimizing therapeutic strategies, or adjusting stimulation in real time, is the lack of established biosensor(s) for vagus nerve engagement and response. The gold standard for assessing nerve engagement is to directly measure the compound action potential (CAP) response to VNS[23,24]. However, CAP recordings (1) are not routinely done in clinical settings, (2) are impractical in small animals, (3) are invasive, and (4) do not provide a readout of central response. As a result, VNS parameter settings are typically adjusted to "the highest 'comfortable' setting" based on patient feedback. This approach impedes principled innovation and likely contributes to the observed large variability in efficacy[3,24]. Noninvasive biosensor(s), which could track the brain's response to VNS, would allow more systematic optimization of treatment in each patient, and potentially open up avenues for closed-loop, adaptive stimulation strategies that respond in real time to the ongoing dynamics of brain activity[4,9].

We and others recently showed that dilation of the pupil at constant luminance provides a sensitive, noninvasive readout of cortical and hippocampal signatures of brain state in mice[25–28]. Pupil dilation tracked basal forebrain cholinergic and locus coeruleus noradrenergic axonal activity in the sensory cortex, with modulator-specific temporal signatures[29]. Pupil dilation also tracks the neuromodulatory brain state in humans and nonhuman primates[30,31]. Here, we test the hypothesis that pupil dilation can serve as a biosensor of VNS-evoked titratable effects on neuromodulatory brain state, as suggested by recent studies[32,33]. Indeed, we find that VNS results in pupil dilation with a magnitude that tracks the stimulation parameters. We further show that this VNS-evoked dilation is mediated by a network involving acetylcholine release from the basal forebrain into the neocortex and interacts nonlinearly with the current momentary state of the brain.

## Results

In order to determine if vagus nerve stimulation (VNS) results in the release of acetylcholine in the cortex and if this VNS-evoked neuromodulation can be tracked using pupillometry, we developed a VNS approach in head-fixed mice. We designed and made custom cuffs for stimulation of the vagus nerve in awake mice, based on our previous design in rats (Fig. 1A, top)[34]. The cuffs were implanted on the left cervical vagus nerve (Fig. 1A, bottom). We stimulated with constant current and measured the return current using a sensing resistor (Fig. 1B). We stimulated with trains of biphasic pulses (to avoid charge accumulation) lasting 10 s (Fig. 1C, top, "Methods"). To reduce anticipation by the mice, the intertrain interval was uniformly distributed between 106 and 130 s. For each train, the pulse amplitude, width, and repetition rate were selected from a set of 60 unique parameter combinations, such that each unique combination was used once per session (Fig. 1C, bottom). In a set of test experiments, we observed that a large fraction of applied current did not return to the negative lead of the cuff (and thus presumably leaked to the ground) when the animal was grounded via the head post (Supplementary Fig. 1A). Therefore, the animal and current stimulation circuit were isolated from the ground in all other experiments (Fig. 1B, bottom). In addition, we found that the narrowest used pulse width (0.1 ms) exhibited modest amplitude reduction (measured online, "Methods"). Furthermore, a small amount of amplitude reduction occurred across all parameter combinations for higher-impedance cuffs, particularly for large-amplitude pulses, resulting from some combination of minor waveform distortion and current pump saturation (Supplementary Fig. 1B–E). These small distortions did not affect the overall results (see below), but suggest that it is beneficial to avoid impedances >15 kOhm (Supplementary Fig. 1F) and pulse widths <0.2 ms.

Mice were head-fixed on a cylindrical treadmill and their pupil size recorded with a custom system ("Methods"). We measured pupil size and exposed eye area from the videos of the animal's eye using DeepLabCut[35] ("Methods"). In some experiments, we simultaneously imaged cholinergic axon activity in the auditory cortex of mice using a two-photon microscope (Fig. 1D). We observed that trains of VNS-elicited pupil dilation (Fig. 1E), cholinergic axon activity in the auditory cortex (Fig. 1F), and probabilistically evoked walking on the treadmill (Fig. 1G). These evoked responses occurred against the backdrop of the ongoing dynamics of these signals across each session (Fig. 1H).

**Graded dependence of pupil dilation on VNS parameters.** To understand the parameter dependence of the VNS-evoked pupil dilation, we first plotted time courses of pupil responses, locked to the time of each VNS train, for each value of each parameter, while averaging across all values of the other two parameters. Pupil dilation had similar time courses across parameter combinations, and the magnitude of dilation increased with increasing

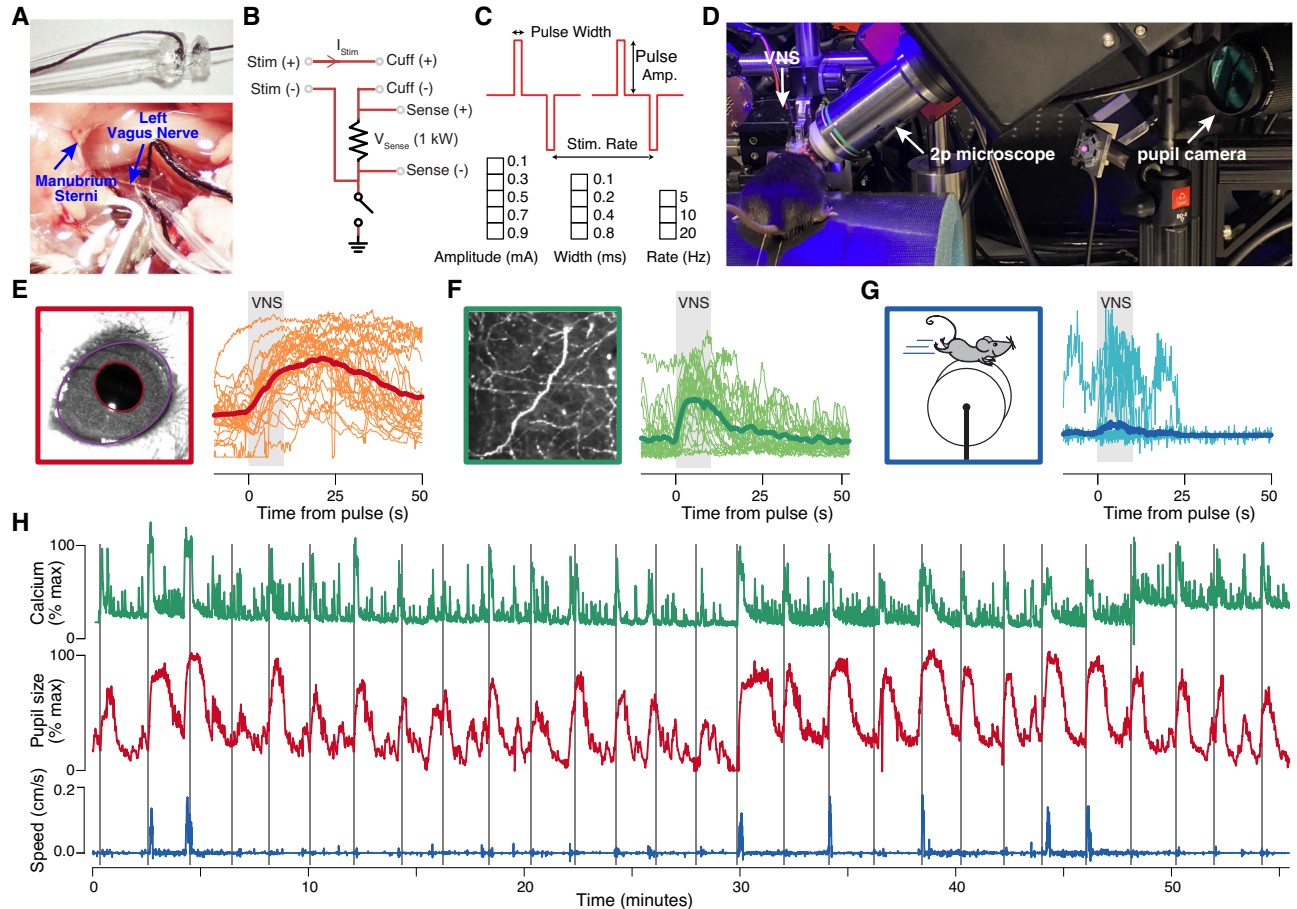

**Fig. 1 Vagus nerve stimulation (VNS), pupillometry, and two-photon axonal GCaMP imaging in the auditory cortex of awake head-fixed mice. A** Bipolar VNS cuff design (top) and implantation on the left cervical vagus nerve (bottom). **B** VNS stimulation circuit diagram. Current is applied between the two cuff electrodes using an improved Howland current pump, and return current is measured across a 1-kOhm sensing resistor. The animal and current pump circuit were isolated from the ground (indicated at the bottom), except where noted. **C** VNS pulse waveforms and parameters. Biphasic pulses, separated by a gap of 50% interpulse interval, were delivered in trains lasting 10 s (top). Five pulse amplitudes, four pulse widths, and three pulse rates were combined to yield 60 unique VNS parameter combinations (bottom). **D** Photograph of a mouse undergoing pupillometry and VNS in the two-photon imaging setup. **E** Left: example video frame of the mouse's eye, with fitted ellipses, overlaid on the pupil (red) and exposed eye area (purple). Right: 60 pupil-size time series (all parameter combinations mentioned in panel **C**) during an example session time-locked to each VNS train (thick line, session mean; thin lines, individual trains). Gray rectangles indicate 10-s window for each train of VNS pulses. **F** Left: example mean fluorescence intensity image from in vivo two-photon GCaMP6 imaging of cholinergic axons in the auditory cortex. Right: example axonal signal time series, aligned as in panel **E**. Same session as in panel **E**. **G** Walking speed time series from the same session as in panels **E** and **F**. **H** Full-session time series of pupil size (red), fluorescence (green), and walking speed (blue) from the same session as in **E**–**G**. Narrow vertical gray rectangles indicate the time of each VNS train.

amplitude (Fig. 2A, left), width (Fig. 2A, middle), or train rate (Fig. 2A, right). We then extracted VNS-evoked pupil response scalars in a window centered on the peak dilation (red bars at the bottom in Fig. 2A; "Methods"). A 3D circle plot of the scalar values showed that the dilation appeared to increase from zero and then saturate (at least partially), for each parameter, and that there were no apparent complex (e.g., nonmultiplicative) interactions between parameters (Fig. 2B). Therefore, we fit a product of log-logistic growth curves ("Methods", Eq. (3)) to the scalar dilations as a function of pulse amplitude, width, and rate (Fig. 2C). The multivariate log-logistic function fits well with the observed marginal distribution patterns. The eyelid receives the same sympathetic innervation as the pupil dilator, but no parasympathetic input[36]. Therefore, we measured the area of the exposed eye as a proxy for the sympathetic component of pupil dilation. Eyelid exhibited a similar pattern of parameter dependence to pupil dilation (Supplementary Fig. 3B–D). The charge at half-maximal pupil dilation (0.15 ± 0.02 μC) was not significantly different from the charge for half-maximal eyelid opening (0.32 ±

0.22 μC, bootstrapped SD, $P = 0.256$), suggesting a prominent sympathetic contribution to the VNS-evoked pupil response.

Because there are large, ongoing fluctuations in pupil size, which reflect changes in brain state (see Fig. 1H)[28], we wondered if the magnitude of the VNS-evoked pupil dilation depended on prestimulation pupil-indexed state. To test this, we conducted a separate series of experiments in which a single VNS parameter combination was used in all 60 repetitions of stimulation for each session ("Methods"). Post hoc sorting of the VNS-evoked pupil responses into bins based on prestimulation pupil size revealed an inverted U dependence of VNS-evoked pupil response on baseline pupil size (Supplementary Fig. 3G).

The visually apparent multiplicative interaction between parameter dependencies, together with the biophysics of extracellular stimulation of axons, suggests that it may be possible to collapse pulse amplitude and width into a single metric: charge/pulse. Indeed, when pupil response was plotted separately for each train rate, as a function of charge/pulse, curves for different pulse widths lied approximately on top of each other (Fig. 3A).

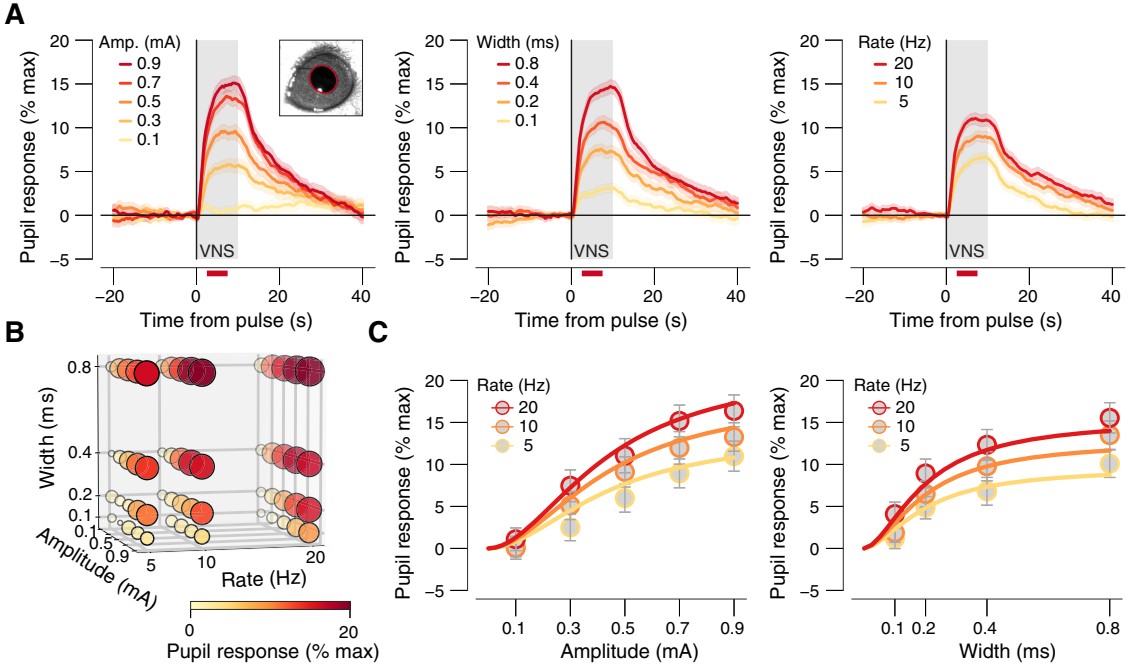

**Fig. 2 Pupil dilation has a graded dependence on VNS parameters. A** VNS-evoked pupil response time courses separately for each pulse amplitude (left), width (middle), and rate (right), collapsed across two other stimulation parameters. Gray window, 10 s VNS train; red bars, the interval for VNS-evoked pupil response scalar measures (see "Methods"); data are presented as mean values ± s.e.m. (across VNS events). **B** VNS-evoked pupil response measure for all 60 unique parameter combinations. Response magnitude is indicated by circle size and color. **C** VNS-evoked pupil response scalars plotted separately for pulse amplitudes and rates (binned across the width, left) and separately for pulse widths and rates (binned across amplitude, right). Colored lines, fitted log-logistic functions ("Methods"); data are presented as mean values ± 1.96 × s.e.m. (across VNS events; see Supplementary Fig. 2 for boxplot representations of the same data separately per animal). All panels: $N = 45$ repetitions for each unique parameter combination (before artifact rejection; "Methods"). Source data are provided as a Source Data file.

Consistent with this visual inspection, a comparable amount of variance was explained when fitting a full hypersurface that depended on amplitude, width, and train rate (26.47%, "Methods", Eq. (3)) versus a reduced surface that only depended on charge/pulse and train rate (26.33%, "Methods", Eq. (4)). A similar dependence on charge/pulse has been observed for VNS-evoked LC firing rates[18]. Therefore, to further reduce the parameter space for subsequent analyses, we defined five charge/pulse "bins" that parse the range of tested values into approximately equal-sized log-spaced charge bins (Fig. 3B). We plotted time courses and scalars sorted by these charge-defined bins (Fig. 3C, D) and assessed the statistical significance of pupil dilation in each charge bin (Fig. 3D, bottom). Dilation was not significant for the smallest charge bin (Fig. 3D, blue area), but was significant for all charge bins above that (Fig. 3D, green and brown areas).

Because of the small size of the mouse neck, and the presence of other nearby nerves, we were concerned that VNS-evoked pupil dilation resulted from the spread of current to other nerves, rather than from activation of the vagus nerve. Therefore, we conducted similar experiments but severed and retracted the vagus nerve above and below the cuff. In this double-cut nerve condition, VNS no longer caused significant pupil dilation, for all but the largest charge/pulse bin (Fig. 3E, F). We therefore defined the largest charge bin as "off target", and the intermediate charge bins as "optimal" (see Fig. 3D, F and connecting arrow). Consistent with these binwise statistics, the charge at half-maximal pupil dilation was significantly larger for the double-cut nerve condition (0.64 ± 0.13 μC) than for intact nerve (0.15 ± 0.02 μC, bootstrapped SD, $P = 0.001$). Similar results were observed if the nerve was cut only above the cuff (proximal to the brain, called single cut; Supplementary Fig. 4B, E), suggesting

that VNS-driven pupil dilation with intact nerve resulted from activation of afferent fibers. When the animal was grounded, substantial dilation occurred during VNS with intact, single- or double-cut nerve (Supplementary Fig. 4G–L), as well as substantial small eye movements phase-locked to the train rate (Supplementary Fig. 5G–L), which was not observed in ungrounded conditions (Supplementary Fig. 5A–F). The charge at half-maximal pupil response on double-cut nerve ungrounded sessions (0.64 ± 0.13 μC) was significantly larger than on double-cut nerve grounded sessions (0.15 ± 0.07 μC, ± bootstrapped SD, $P = 0.005$). Thus, if care is taken to prevent current spread to the ground, and charge/pulse is kept below ~0.3 μC, pupil dilation provides a sensitive readout of the strength of brain activation resulting selectively from the firing of fibers in the vagus nerve bundle.

**VNS activates cholinergic axons in the auditory cortex.** Having demonstrated that VNS results in dilation of the pupil, we wondered if VNS evokes acetylcholine release from the basal forebrain into the neocortex. We previously showed that long-lasting (e.g., >2–3 s) pupil dilation closely tracks cholinergic axonal activity in the auditory and visual cortex in similar behavioral conditions to those used in this study[29]. Therefore, we suspected that VNS-evoked pupil dilation was linked to the release of acetylcholine from the basal forebrain into the cortex (BF-ACh). To test this hypothesis, we imaged populations of BF-ACh axons in the auditory cortex in mice conditionally expressing GCaMP6s in cholinergic neurons ("Methods"). We observed a dense plexus of axons in layer I of the auditory cortex, consistent with the known anatomical innervation pattern (Fig. 4A, left). To ask how activity in these axons was linked to VNS

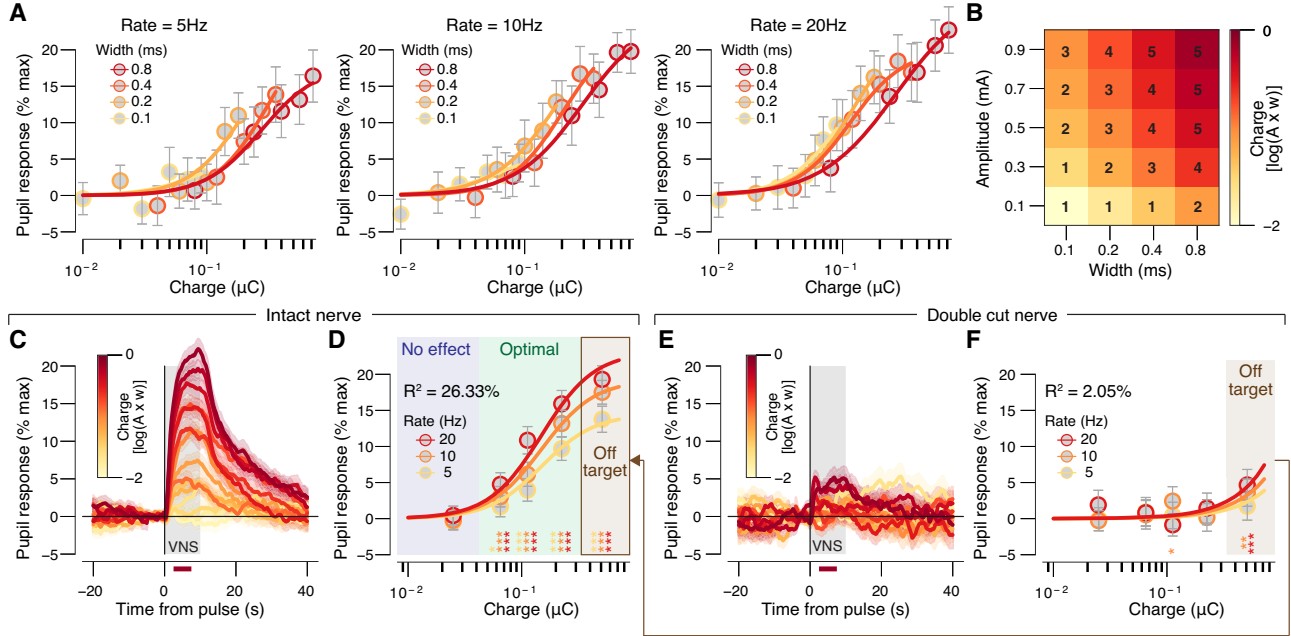

**Fig. 3 VNS-evoked pupil dilation requires an intact nerve and localized current. A** VNS-evoked pupil response measures separately per pulse-charge (amplitude × width) and width, for 5 Hz trains (left), 10 Hz trains (middle), and 20 Hz trains (right). Colored lines, fitted log-logistic function (see "Methods"); data are presented as mean values ± 1.96 × s.e.m. (across VNS events). **B** Grid showing the width and amplitude parameter combinations that were collapsed into our 5 charge/pulse bins. **C** VNS-evoked pupil time courses, each trace is a charge/pulse bin and train rate. Gray window, VNS train; red bar, the interval for VNS-evoked pupil response scalar measures (see "Methods"); data are presented as mean values ± s.e.m. (across VNS events). **D** VNS-evoked pupil response measures separately per charge/pulse bin and train rate. Colored lines, fitted log-logistic function ("Methods"); data are presented as mean values ± 1.96 × s.e.m. (across VNS events); stats, two-sided one-sample t test (tested against 0; ***P < 0.001, **P < 0.01; *P < 0.05, corrected for false discovery rate); blue, green, and brown backgrounds indicate no effects, optimal, and off-target pulse charges, respectively ("Methods"). **E**, **F** As **C**, **D**, but for double-cut nerve, ungrounded sessions. **A–D** N = 45 repetitions for each unique parameter combination (before artifact rejection; "Methods"). **E**, **F** N = 24 repetitions for each unique parameter combination (before artifact rejection, "Methods"). Source data are provided as a Source Data file.

stimulation, we first calculated the cluster-corrected pixelwise significance of the BF-ACh axon response to VNS (irrespective of VNS parameters). This approach provides an unbiased search for significant pixels, similar to what is commonly done with fMRI imaging[37]. We found that the significant pixels mapped closely to a large majority of the axonal plexus (Fig. 4A, right), and that this activity was not a result of brain or skull motion (Supplementary Fig. 6A–C).

To explore the parameter dependence of VNS-evoked BF-ACh axon activity, we tested all combinations of a reduced parameter set compared to that used with the full exploration of pupil dilation (see Supplementary Fig. 6G). This was done in order to keep the imaging sessions within 1 h and avoid excessive bleaching and/or photodamage to the tissue. We found that the VNS-evoked BF-ACh axon activity exhibited a similar time course (Fig. 4B) and parameter dependence (Fig. 4B–F) to that of pupil dilation seen above (and see Supplementary Fig. 6D–F). As seen above for pupil responses, a comparable amount of variance was explained when fitting a full hypersurface that depended on amplitude, width, and train rate (49.11%, "Methods", Eq. (3)) versus a reduced surface that only depended on charge/pulse and train rate (48.89%, "Methods", Eq. (4)). The significant axonal activity was observed in the same optimal zone of parameter space and saturated before entering the off-target zone (Fig. 4F). To avoid "double dipping", binwise significance was tested by comparing the VNS-evoked calcium responses in each of the highest four charge bins to responses in the lowest charge bin, rather than to zero ("Methods"). Consistent with these binwise statistics, the charge at half-maximal pupil dilation (0.06 ± 0.01 μC) was not significantly different from the charge for half-

maximal calcium response (0.062 ± 0.01 μC, bootstrapped SD, P = 0.17). Thus, parametrically varied VNS results in the titratable release of acetylcholine in the neocortex that is well-tracked by pupil dilation.

**Graded dependence of locomotion on VNS parameters.** In addition to causing pupil dilation, we observed that animals sometimes walked on the treadmill during and shortly after trains of VNS. We therefore analyzed the probability of walking as a function of the VNS parameters. Evoked walkbouts occurred against a backdrop of intermittent spontaneous walking (Supplementary Fig. 7B–E), which we corrected for in subsequent analyses (see "Methods"). We defined walking as an average velocity during VNS of >0.05 cm/s (Supplementary Fig. 7A). Even with this low threshold velocity threshold, walking occurred on a substantial fraction of trials only for fairly strong stimulation and reached ~40% of trials in the off-target stimulation zone (Fig. 5A, B). The charge at half-maximal pupil response (0.15 ± 0.02 μC) was not significantly different from the charge at half-maximal walking probability (0.17 ± 0.05 μC, bootstrapped SD, P = 0.35), suggesting a shared underlying driving circuit.

To further characterize the VNS-evoked walking, we measured the distance traveled time courses for those trials with significant walking. When mice walked, they did so mostly during the stimulation and stopped shortly afterward, and the distance traveled appeared weakly parameter-dependent (Fig. 5C–E). To quantify the parameter dependence, we calculated the mean speed during the 10 s of VNS for those trains that elicited significant dilation. This average walking speed during the 10 s trains

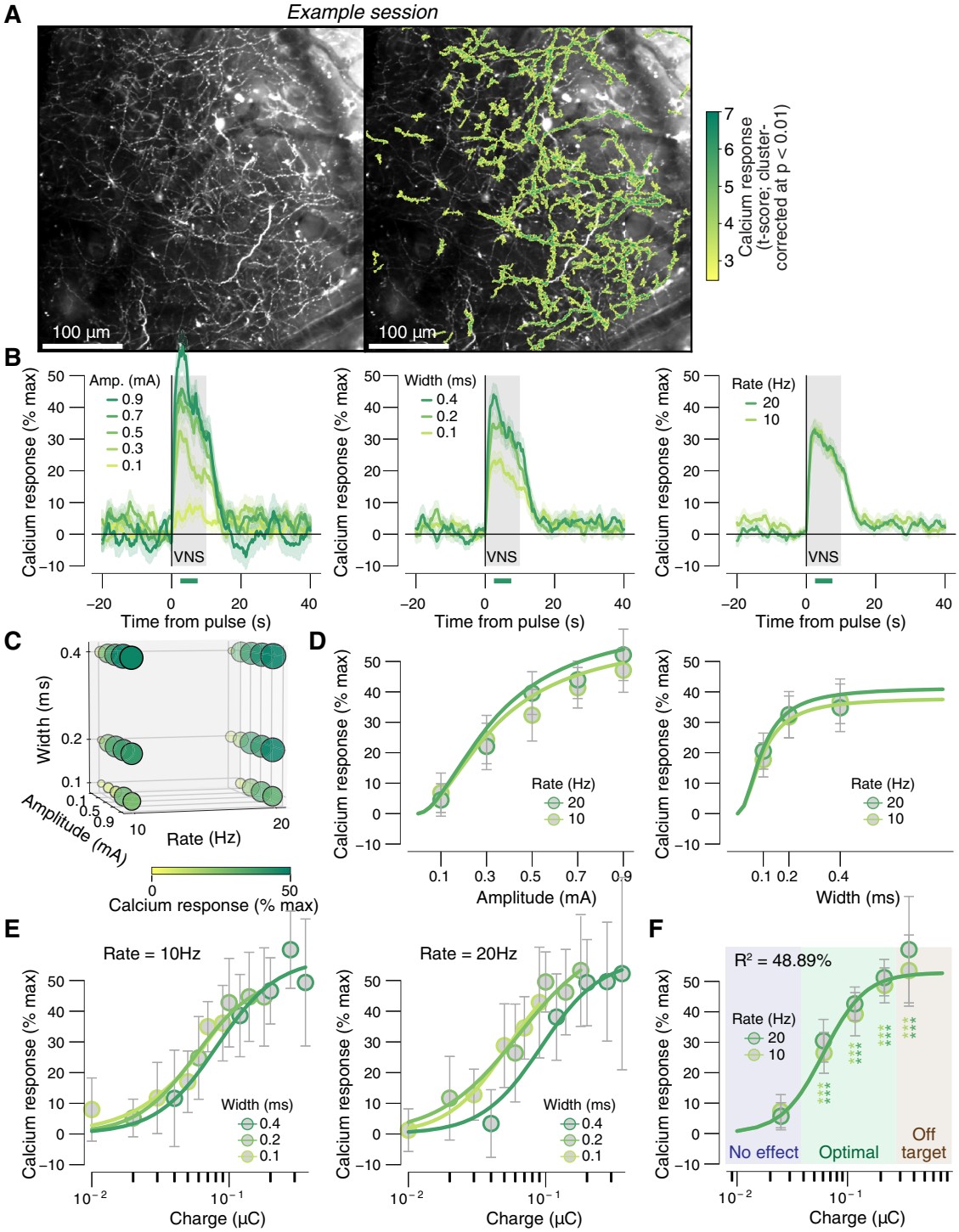

depended weakly on pulse amplitude (Fig. 5D, left) and even less so on pulse width (Fig. 5D, right) or stim rate (Fig. 5D, E). Thus, when walking did occur, it manifested as a fairly stereotyped walk-bout at 0.3–0.4 cm/s lasting the duration of the VNS, followed by slowing to a stop. We did not observe VNS-evoked changes in movement in the double-cut experiments in all but the highest charge bin (Supplementary Fig. 7G, I). This pattern mimics VNS-evoked pupil responses (Fig. 3F), suggesting that indeed there is no off-target driving of locomotor activity or pupil dilation, except for those very large charges which we consistently show have off-target effects with intact nerve, as well.

**Mediation analysis of VNS effects on pupil-indexed neuromodulation.** In our BF-ACh imaging experiments, we had measurements of VNS stimulus parameters and VNS-evoked axonal activity, pupil dilation, and walking patterns. We sought to understand the interdependence of these four variables. First, we excluded VNS trains with walking (Supplementary Fig. 8A) and plotted the calcium response and pupil dilation as a function of charge/pulse (Fig. 6A, B). We found that axonal response (Fig. 6A) and pupil dilation (Fig. 6B) exhibited a similar parameter dependence to that observed previously (without excluding walking trials). Therefore, we concluded that VNS-evoked

**Fig. 4 Graded VNS causes graded neuromodulation recruitment. A** Left: example session mean fluorescence of cholinergic axons in layer 1 of the auditory cortex. Right: as left, but with an overlaid color map of VNS-evoked calcium responses (across all VNS stimulations, see "Methods"). This map was used as a session-wise region of interest for the analysis of calcium signal parameter dependence. This is a representative example of 12 independent repetitions (sessions). **B** VNS-evoked calcium response measures ("Methods") for all 30 unique parameter combinations. Response magnitude is indicated by circle size and color. **C** VNS-evoked calcium responses separately for pulse amplitudes (left), widths (middle), and rates (right) collapsed across two other stimulation parameters. Gray window, 10 s VNS train; green bar at the bottom, the interval for VNS-evoked axonal calcium response measures ("Methods"); data are presented as mean values ± s.e.m. (across VNS events; see Supplementary Fig. 6J, K for boxplot representations of the same data separately per animal). **D** VNS-evoked calcium response measures separately for pulse amplitudes and rates (left, collapsed across widths) and separately for pulse widths and rates (right, collapsed across amplitudes). Colored lines, fitted log-logistic function ("Methods"); data are presented as mean values ± 1.96 × s.e.m. (across VNS events). **E** VNS-evoked calcium response measures separately per pulse-charge (amplitude × width) and width, 10 Hz trains (left), and 20 Hz trains (right). Colored lines, fitted log-logistic function (see "Methods"); data are presented as mean values ± 1.96 × s.e.m. (across VNS events). **F** As in **D**, but for charge/pulse bins. Stats, two-sided paired-samples $t$ test (tested against the response magnitudes in the lowest charge bin; ***$P < 0.001$, **$P < 0.01$; *$P < 0.05$ false discovery rate corrected); blue, green, and brown backgrounds indicate no effects, optimal, and off-target pulse charges, respectively, defined by the pupil response data (as in Fig. 3D, F; "Methods"). **B**–**D** $N = 12$ repetitions for each unique parameter combination (before artifact rejection; "Methods"). Source data are provided as a Source Data file.

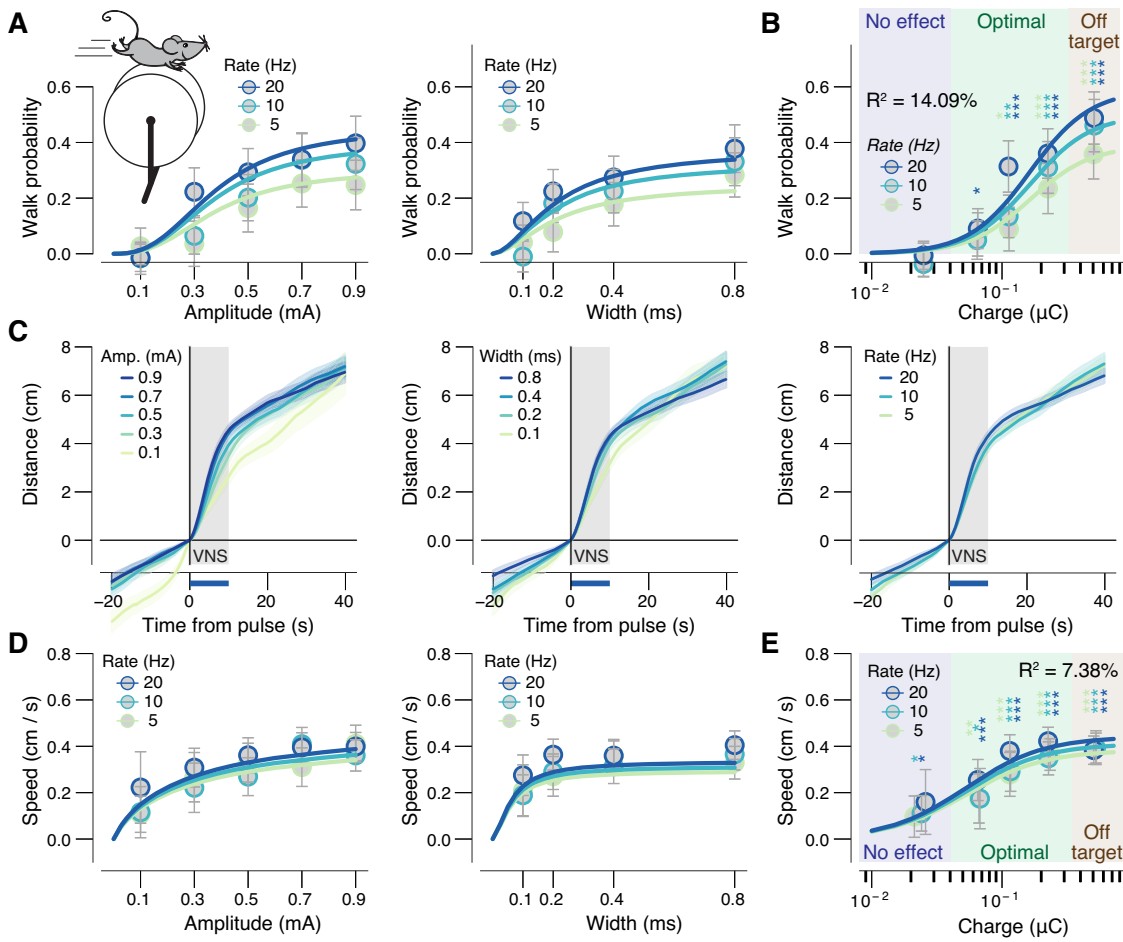

**Fig. 5 VNS probabilistically causes locomotor activity. A** VNS-evoked walk probability (see "Methods") separately for pulse amplitudes and rates (collapsed across widths, left) and separately for pulse widths and rates (collapsed across amplitudes, right). Colored lines, fitted log-logistic function ("Methods"); data are presented as mean values ± 1.96 × s.e.m. (across VNS events). **B** As **A**, but for charge bins. Stats, two-sided one-sample $t$ test (tested against 0; ***$P < 0.001$, **$P < 0.01$; *$P < 0.05$ false discovery rate corrected); blue, green, and brown backgrounds indicate no effects, optimal, and off-target pulse charges, respectively, defined by the pupil response data (as in Fig. 3D, F; "Methods"). **C** Walked distance (walk trials only) after VNS train start, separately for pulse amplitudes (left), widths (middle), and rates (right) collapsed across two other stimulation parameters. Gray window, pulse duration; blue bar, the interval for VNS-evoked walking speed measures ("Methods"); data are presented as mean values ± s.e.m. (across VNS events). **D**, **E** As **A** and **B**, but for walking speed on walk trials only. All panels: $N = 27$ repetitions for each unique parameter combination (before artefact rejection; "Methods"). Source data are provided as a Source Data file.

pupil-indexed neuromodulation did not depend on locomotor activity. Having removed walking from the picture, we then compared the VNS train-wise magnitudes of pupil response and calcium response. We found VNS-evoked pupil dilation and BF-

ACh axonal activity were robustly correlated (Fig. 6C), even after regressing out their dependence on VNS parameters (Fig. 6D).

To further characterize the effects of VNS on ACh release and pupil dilation, we conducted a mediation analysis, asking to what

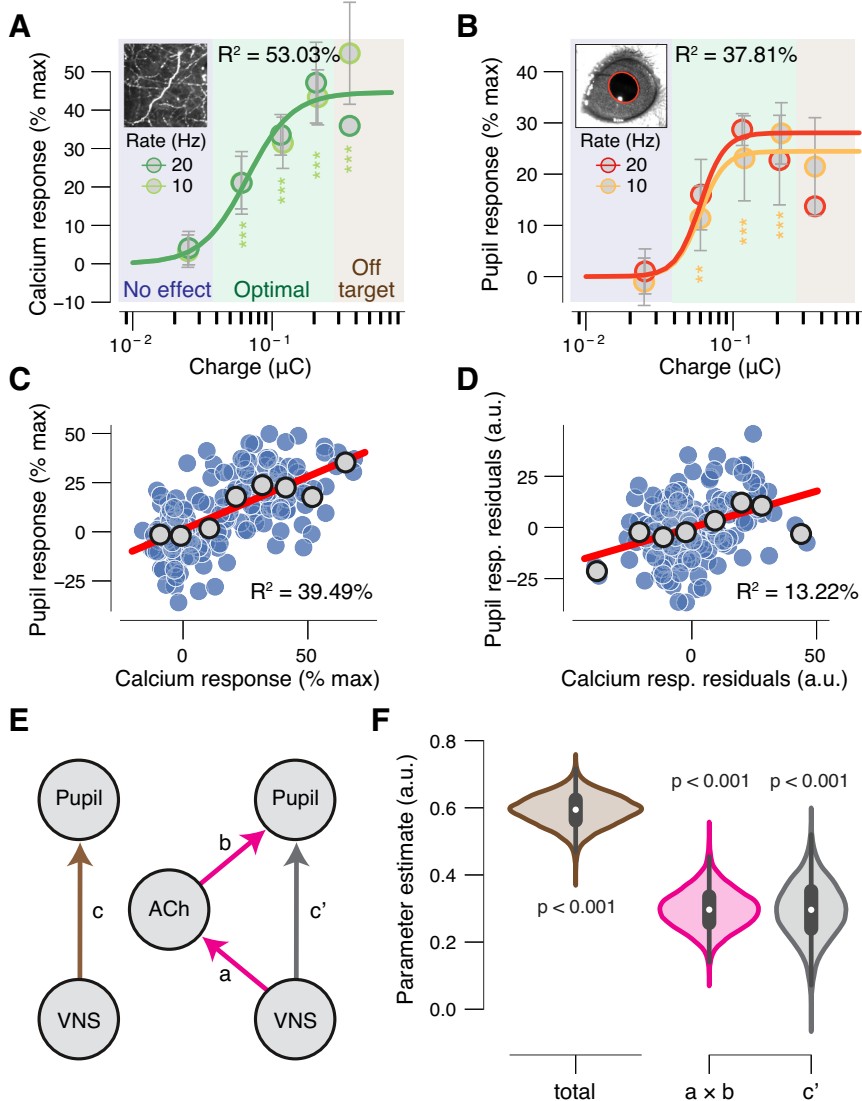

**Fig. 6 VNS triggered cortical acetylcholine release mediates a large fraction of pupil dilation. A** VNS-evoked calcium response measures, in absence of walking, separately per charge/pulse bin and train rate. Colored lines, fitted log-logistic function ("Methods"); data are presented as mean values ± 1.96 × s. e.m. (across VNS events); stats, two-sided one-sample $t$ test (tested against 0; ***$P < 0.001$, **$P < 0.01$; *$P < 0.05$ false discovery rate corrected); blue, green, and brown backgrounds indicate no effects, optimal and off-target pulse charges, respectively, defined by the pupil response data (as in Fig. 3D, F; "Methods"). **B** As **A**, but for VNS-evoked pupil responses during the imaging sessions. **C** Scatterplot of the relationship between VNS-evoked pupil responses and VNS-evoked calcium responses, in absence of walking. Blue data points are individual VNS trains; gray data points are calcium response defined bins (equal size); red line, linear fit; Pearson correlation, $P < 0.001$. A first-order (linear) fit was superior to a constant fit ($F_{1,178} = 116.19$, $P < 0.001$), and a second-order (quadratic) fit was not superior to the first-order fit ($F_{1,178} = 1.2$, $P = 0.28$) (sequential polynomial regression; "Methods"). **D** As **C**, but after removing the effect of VNS (partial correlation). Pearson correlation, $P < 0.001$. A first-order (linear) fit was superior to a constant fit ($F_{1,178} = 26.13$, $P < 0.001$), and a second-order (quadratic) fit was not superior to the first-order fit ($F_{1,178} = 0.08$, $P = 0.78$). **E** Left, schematic of relationship between VNS and pupil responses. Arrow, regressions; coefficient $c$ quantifies the "total effect". Right, schematic of mediation analysis of VNS to pupil responses, via cholinergic axons responses ("Methods"). Arrows, regressions; coefficient $a \times b$ quantifies the "indirect" (mediation) effect; coefficient $c'$ quantifies the "direct effect". **F** Fitted regression coefficients (kernel density estimate of 5 K bootstrapped replicates) of the total effect (brown), indirect path (mediation; pink), and the direct path (gray). Stats, the fraction of bootstrapped coefficients smaller than 0. All panels: $N = 12$ repetitions for each unique parameter combination (before artefact rejection; "Methods"). Source data are provided as a Source Data file.

extent the recruitment of ACh statistically mediates the apparent effect of VNS on pupil dilation (Fig. 6E). We found that the indirect path of VNS driving ACh, in turn driving pupil dilation, partially mediates the apparent effect of VNS on the pupil (Fig. 6F; 49.7% of the total effect). Similar results of the mediation analysis were observed if walking trials were included (Supplementary Fig. 8D–F). Taken together, we conclude that pupil dilation provides a sensitive readout of VNS-evoked cholinergic modulation of the cortex.

## Discussion

VNS is a widely used clinical therapy and is thought to exert its effects by changing the neuromodulatory state of the brain. However, it has not been clear which neuromodulators are evoked by VNS. Furthermore, because there has not been an established biosensor to titrate stimulation on a patient-by-patient basis, or in real time, optimizations of VNS parameters are made only coarsely, often across months or years based on patient feedback. Here, we have demonstrated that pupil dilation

is a sensitive readout of VNS strength and of the extent of evoked release of acetylcholine in the sensory cortex. Furthermore, using single- and double-cut nerve experiments, together with careful measurement of impedance and current leakage, we have defined the space of optimal stimulus parameters and grounding conditions to selectively recruit the vagus nerve bundle and alter cortical brain state. Our approach can be applied to use VNS to achieve titratable cortical neuromodulation in closed-loop frameworks with pupil dilation as a feedback signal for a wide range of basic neuroscience and clinical translational models.

A major impediment to parsing mechanisms, optimizing therapeutic strategies, or adjusting stimulation in real time, is the lack of established biosensors for successful VNS. Our results establish pupil dilation as a reliable biosensor, indicated by nerve engagement without off-target effects. This result could be especially useful during "transcutaneous" VNS (t-VNS)[38], which involves electrically stimulating the nerve's cervical or auricular branches close to the surface of the skin. However, it was recently reported that t-VNS through its auricular branch did not affect pupil size[39–41]. The discrepancy between these findings and ours suggest that the transcutaneous protocol in those studies likely insufficiently stimulated the vagus nerve, perhaps due to the site of stimulation (auricular vs cervical branches), the chosen stimulation parameters (pulse amplitude, width, rate), or stimulation duration. Indeed, it was recently shown that shorter tVNS pulses than previously used did result in reliable pupil dilation[42]. Taken together, these results suggest that future tVNS approaches that increase nerve engagement may hold promise for noninvasive and low-risk (closed-loop) VNS with pupillometry.

Our results are important for the mechanistic interpretation of pupil size measurements, a technique whose use is growing exponentially[43]. Fluctuations in pupil size at constant luminance are widely used as a noninvasive peripheral readout of changes in cortical neuromodulatory arousal state[28,29,44]. Changes in pupil diameter are commonly associated with locus coeruleus (LC) responses in humans[31,45], monkeys[30,46], and mice[29,47]. Our results add to a growing appreciation that pupil responses track not only activity of the LC, but also that of other subcortical regions, including the dopaminergic midbrain, the tectum, and indeed the cholinergic basal forebrain from mice to humans[29–31,48].

The finding that VNS-evoked pupil responses and baseline pupil size interact in a non-monotonic (inverted U) fashion, albeit weakly, indicates that the largest VNS-evoked responses of cholinergic axons likewise occurred for intermediate levels of tonic arousal. This result supports the model that phasic neuromodulatory signals that occur naturally are strongest during intermediate levels of tonic neuromodulatory activity[49]. This model is further supported by recent observations that tonic arousal predicts non-monotonic (inverted U) changes in the signal-to-noise ratio of sensory cortical responses and perceptual sensitivity[25,50,51]. Importantly, we observed the inverted U relationship after correcting for a general reversion to mean. Reversion to the mean predicts the commonly observed linear anti-correlation between baseline pupil size and task-evoked responses[52,53]. We believe this mean reversion is highly underappreciated in the analysis of pupil responses, in general, and may merit reinterpretation of a large literature on pupil dependencies of physiology and behavior.

In addition to pupil dilation, we also observed walking in response to VNS on some trials, particularly with high shock strengths. However, we believe pupil dilation is the more useful biomarker than locomotion for a number of reasons. First, our finding that VNS-evoked pupil responses occur even in the absence of walking suggests that pupil dilation is the more sensitive readout. Second, walking occurred only probabilistically,

and at higher shock strengths. Third, humans experience voluntary control over walking, while pupil dilation is largely involuntary. Fourth, pupil size measurements require nothing more than constant luminance levels and a relatively still head, so there are fewer restrictions on the experimental setup. In contrast, measuring walking requires a treadmill, which is particularly problematic in combination with human functional neuroimaging.

Although the extent of cholinergic axon activation was highly correlated to VNS stimulation intensity in our data, the precise brain circuit linking VNS to cortical acetylcholine is far from clear. The main effects of VNS on the brain are thought to be mediated by vagus nerve sensory afferents activating the nucleus tractus solitarii (NTS), which in turn activates the reticular activating system and other brain areas[54]. However, it is currently unclear in our study and others which of the three major fiber types (A, B, and/or C) may be stimulated and play a role[34]. In addition, other fiber bundles, such as from the superior cervical ganglion (SCG) sometimes travel with the vagus nerve[55,56], and thus may contribute to cortical effects and/or pupil dilation. Four aspects of our results point to the vagus/NTS-mediated circuit: (1) no pupil response after cutting the vagus nerve, (2) our control experiment on grounded conditions show that there was virtually no current leakage in our experimental conditions, (3) the strong correlation of VNS shock strength to cortical acetylcholine, and (4) the tendency of this correlation to formally mediate the effects of VNS on pupil response magnitude. Beyond NTS, the circuits driving pupil and cortical acetylcholine are also unknown. VNS elicits activity in many brain regions, including the amygdala, thalamus, hypothalamus, and cerebral cortex[14,15], perhaps through neuromodulators, such as NE from LC. A prominent role for NE-LC is supported by lesion studies[16] and the parameter-dependent recruitment of LC activity with VNS[17,18]. NE-LC may thus play a driving role both in the pupil dilation and of the basal forebrain acetylcholine release we observed. However, extensive future work will be needed to pinpoint the circuit linking VNS to cortical state and pupil size.

Acetylcholine is heavily involved in normal executive and mnemonic functioning, and the loss of cholinergic signaling and cognitive decline are closely linked[57–59]. Our results suggest that the (therapeutic) benefits of VNS may be at least partially mediated by the recruitment of the basal forebrain acetylcholine system. Exploring this hypothesis, using the genetic tools for dissecting and perturbing neural circuits at the micro- and meso-levels in mouse models, holds promise in various disease models, such as Alzheimer's[3,13,60]. As VNS is increasingly applied to a broad range of brain disorders, validation of biosensors, elucidation of mechanisms, and optimization of stimulation techniques will be integral components of achieving success. Our results demonstrate that pupil dilation can be used as a biosensor, that cortical release of acetylcholine is a component mechanism, and that control of impedance, current leakage, and stimulus parameters provide important ingredients for the improved use of VNS.

## Methods

**Animals**. All surgical and animal handling procedures were carried out in accordance with the ethical guidelines of the National Institutes of Health and were approved by the Institutional Animal Care and Use Committee (IACUC) of Baylor College of Medicine. We used a cross of two transgenic lines for the imaging experiments, and wild-type mice in all other experiments. The wild-type mice ($N = 31$; all male) were of C57BL/6 strain (Jackson Labs). The mice for cholinergic axon imaging ($N = 3$; 2 female; age, 8–18 weeks) were heterozygous for the ChAT-cre (Jackson Labs strain B6; 129S6-Chattm1(cre)Lowl/J) and Ai162 GCaMP6 reporter lines (Allen Institute). Mice received ad libitum food and water and were individually housed after VNS cuff implantation. Mice were kept on a regular light-dark cycle, and all experimental manipulations were done during the light phase.

**Table 1 Sample sizes in the parameter exploration experiment.**

|  | Intact nerve | Single-cut nerve | Double-cut nerve |
|---|---|---|---|
| Ungrounded animal | 10 animals | 3 animals | 5 animals |
|  | 45 sessions | 14 sessions | 24 sessions |
|  | 2700 VNS events | 840 VNS events | 120 VNS events |
| Grounded animal | 7 animals | 3 animals | 3 animals |
|  | 22 sessions | 17 sessions | 14 sessions |
|  | 1320 VNS events | 1020 VNS events | 840 VNS events |

Format: total number of animals, sessions, and VNS events.

**Table 2 Sample sizes in rest of experiments.**

| Baseline pupil-dependence experiment | Axon imaging experiment | Light control experiment |
|---|---|---|
| 4 animals | 3 animals | 1 animal |
| 23 sessions | 12 sessions | 2 sessions |
| 1380 VNS events | 360 VNS events | 120 VNS events |

Format: total number of animals, sessions, and VNS events.

**Sample sizes**. See Tables 1 and 2 for sample sizes of each experiment. Some animals in the "parameter exploration experiment" underwent both grounded and ungrounded sessions. In the "baseline pupil-dependence experiment", we excluded 8 (out of 23) sessions in which the mean VNS-evoked pupil response was smaller than 1% signal change. For the "axon imaging experiment", we performed a total of 31 imaging sessions in three unique mice. Of these, 12 imaging sessions were successful (three, four, and five sessions per mouse, respectively). We excluded the remaining sessions from the analyses based on three criteria: (i) no observable axons after preprocessing (4 sessions), (ii) more than 2 µm image motion on more than ten trials (see also "Analysis of calcium imaging data", below) (one session), and (iii) overall VNS-evoked activity in less than 750 pixels (see also "Analysis of calcium imaging data", below) (19 sessions).

**Vagus nerve stimulator cuff design**. We adapted the implanted cuff design from prior work in rats[34]. The bipolar stimulating cuff was custom built based around an improved Howland Current Pump (Texas Instruments) driven by an 18-bit NI DAQ (National Instruments). Two platinum–iridium wires were fixed 1 mm apart to biocompatible micro silicone tube (cuff: 0.5 mm inner diameter, 1.0 mm outer diameter, 2.0 mm long; lead: 0.3 mm inner diameter, 0.6 mm outer diameter, 30 mm long). The end of the two lead wires was connected to gold pins (363 A/PKG, Plastic One), which were used to connect the cuff to the stimulator.

**Surgical procedures**. The surgical station and instruments were sterilized prior to each surgical procedure. Isoflurane anesthetic gas (2–3% in oxygen) was used for the entire duration of all surgeries. The temperature of the mouse was maintained between 36.5 and 37.5 °C using a homeothermic blanket system.

*Head-post implantation*. After anesthetic induction, the mouse was placed in the stereotax. The surgical site was shaved and cleaned with scrubs of betadine and alcohol. A 1.5–2 cm incision was made along the scalp midline, the scalp and overlying fascia are retracted from the skull. A sterile head post was then implanted using dental cement.

*Cervical vagus nerve cuff electrode implantation*. Following induction, the mouse was placed in a supine position; the surgical site was shaved and cleaned with scrubs of betadine and alcohol. A 1–1.5 cm long midline incision was made from manubrium to jawline. The blunt dissection technique was used for the whole procedure. With tiny blunt-tipped scissors, the left submaxillary gland was separated from the connective tissue and retracted to another side. The connective tissue between the left sternocleidomastoid and the sternohyoid /omohyoid was carefully separated until the carotid sheath was visible. Using a small pair of surgical retractors to hold the muscles apart, a 4–5 mm segment of the left cervical vagus nerve was dissected from the carotid sheath and the cuff electrode was positioned around the vagus nerve. We ensured that the electrode wires had circumferential or near-circumferential contact with the nerve. The cuff was fastened with silk suture. The muscles were placed back in their original position and an absorbable suture was used to secure the cuff in position. A subcutaneous tunnel was made in between the ear and eyes from neck incision to the top of the head, which allowed passing the cuff leads to the skull. Lead pins were fixed to the previously implanted head post using acrylic dental cement. The submaxillary gland was placed back to its original position and the incision was sutured. While

still under anesthesia, VNS response was recorded using a pulse-oximeter (Starr Life Sciencvs, MouseOx Plus), a reduction in blood oxygen saturation was observed when stimulation was given through the implanted cuff, confirming that the electrode was appropriately positioned to stimulate the intact and functioning vagus nerve.

*Craniotomy*. We made a cranial window over the right auditory cortex[25,29]. We removed skin over the skull and bluntly dissected part of the temporalis muscle off to determine the location of the craniotomy. A 3-mm window was opened using a handheld dental drill (Osada Exl-M-40). We occasionally applied gel foam soaked in saline to the dura mater to stop any small bleeding. The dura was then covered with a thin layer of silicon oil and a coverslip was laid over the window and sealed with surgical glue (Vetbond, 3 M). We applied dental acrylic cement throughout the exposed skull surface as well as across a small rim of the coverslip to secure it.

**Vagus nerve stimulation protocol**. Custom LabView software handled data acquisition and synchronization with VNS. The VNS leads were connected to the output of an enhanced Howland Current Pump (HCP) with a current scaling factor of 1 mA/V. A data acquisition board (National Instruments USB-6353 X Series) was used to interface with a computer running MATLAB R2016. Vagus nerve stimulation via our cuff was well-tolerated by our mice for up to several months. Cuff impedance was monitored before most sessions using an electrode impedance tester (BAK Electronics, INC: IMP-2).

Animals were allowed to recover for 2–3 days prior to recordings. The animal's head was fixed, and they were allowed to spontaneously walk or stand still on a cylindrical treadmill along a single axis of rotation. In all experiments, the intertrain interval was uniformly distributed between 106 and 130 s (mean, 118 s; varied to reduce expectancy), and each pulse train lasted for 10 s. We performed four different types of experiments: First, in the "parameter exploration" experiment (Figs. 2, 3, and 5), we performed an exhaustive search across a three-dimensional VNS parameter space: five pulse amplitudes (0.1, 0.3, 0.5, 0.7, or 0.9 mA), four pulse widths (0.1, 0.2, 0.4, or 0.8 ms), and three pulse rates (5, 10, or 20 Hz). Each unique parameter combination (60 in total) occurred once in a session of 2 h. Second, in the "baseline pupil-dependence experiment" (Supplementary Fig. 3E–G), we administered 60 pulses with the same combination of parameters in each session: amplitude, 0.5 or 0.7 mA; width, 0.2 ms; rate, 10 or 20 Hz rate. Third, in the "axon imaging experiment" (Figs. 4 and 6), we performed an exhaustive search across a subset of the three-dimensional VNS parameter space: five pulse amplitudes (0.1, 0.3, 0.5, 0.7, or 0.9 mA), three pulse widths (0.1, 0.2, or 0.4 ms) and two pulse rates (10 or 20 Hz). Each unique parameter combination (30 in total) occurred once in a session of 1 h. Fourth, the "light control experiment" (Supplementary Fig. 6I), was the same as the axon imaging experiment (performed on the microscope setup but without actual imaging), except for that we increased the background luminance in the imaging setup to our standard level in the non-imaging setups.

*Grounded vs ungrounded sessions*. In all experiments (except where noted), to unground the animals, we put insulating tape between the head-post holder and other metal parts connected to it. In grounded animal experiments, a wire connected the head-post holder to the ground. To assay the effects of grounding on cuff performance, the applied current was compared to the measured return current, which was measured across a current-sensing resistor.

**Acquisition of pupil, exposed eye area, and walking data**. We continuously recorded the eye contralateral to the cuff implantation with a Basler GigE camera (acA780-75gm), coupled with a fixed focal length lens (55 mm EFL, f/2.8, for 2/3"; Computar) and infrared filter (780-nm long pass; Midopt, BN810-43), positioned ~8 inches from the mouse. An off-axis infrared light source (two infrared LEDs; 850 nm, Digikey; adjustable in intensity and position) was used to yield a high-quality image of the surface of the eye and a dark pupil. Images (504 × 500 pixels) were collected at 15 Hz during the imaging experiment, and at 143 Hz for all other experiments, using a National Instruments PCIe-8233 GigE vision frame grabber. To achieve a wide dynamic range of pupil fluctuations, an additional near-ultraviolet LED (405–410 nm) was positioned above the animal and provided low-intensity illumination that was adjusted such that the animal's pupil was approximately mid-range in diameter following placement of the animal in the setup, and did not saturate the eye when the animal walked. Near-ultraviolet LED light levels were lower during two-photon imaging experiments, to avoid noise on the photo-multiplier tubes. We continuously measured treadmill motion using a rotary optical encoder (Accu, SL# 2204490) with a resolution of 8000 counts/revolution.

**Acquisition of cholinergic axon data (in vivo calcium imaging)**. We performed in vivo GCaMP imaging of cholinergic axons in the right auditory cortex with a fast resonant scanning system (ThorLabs rotating Bergamo). The imaging frame rate was 15 Hz. Excitation was via a Ti–sapphire laser (Insight DS+, Spectra Physics) tuned to 930 nm, with either a ×16 (0.8 NA, Nikon) or ×25 (1.1 NA, Nikon) objective at a depth of 100–200 µm from pial surface. Power out of the objective was controlled by calibrated rotations of a half-wave attenuator and depended on

the magnification of the scan but was typically 30–40 mW. We used Scan Image (Vidrio) to control the imaging system. The imaging data was synchronized to the VNS timings, videos of the pupil, and wheel motion via custom Labview software.

**Analysis of leak fraction and filtering**. We characterized the extent of current leakage and filtering (Supplementary Fig. 1) as follows. First, leak fraction was defined as:

$$L = 1 - \frac{A_m}{A_i} \tag{1}$$

where $L$ was the session-wise leak fraction, $A_m$ was the mean across the minimum measured amplitude for each unique pulse widths and rate (to avoid filtering effects (see below), we excluded the 0.1 and 0.2 ms pulse widths because of the filter as described next), and $A_i$ was the session-wise minimum intended amplitude (0.1 mA in the ungrounded sessions; 0.1, 0.2, 0.4, or 0.6 mA in the grounded sessions). Second, the extent of filtering was modeled with the following sigmoidal function:

$$A_m = \frac{s}{1 + \exp(-a(W - b))} \tag{2}$$

where $A_m$ was a vector of the mean measured amplitudes for each pulse width, $W$ was a vector of the unique pulse widths (0.1, 0.2, 0.4, and 0.8 ms), and $s$, $a$, and $b$ were the free parameters of the fit.

**Analysis of pupil and eyelid data**. All analyses were performed using custom-made Python scripts, unless stated otherwise.

*Preprocessing.* We measured pupil size and exposed eye area from the videos of the animal's eye using DeepLabCut[35]. In approximately 1000 training frames randomly sampled across all sessions, we manually identified eight points spaced approximately evenly around the pupil, and eight points evenly spaced around the eyelids (Supplementary Fig. 3A). The network (resnet 110) was trained with default parameters. In order to increase the network's speed and accuracy when labeling (unseen) frames of all videos, we specified video-wise cropping values in the DeepLabCut configuration file that corresponded to a square around the eye. The pupil size (exposed eye area) was computed as the area of an ellipse fitted to the detected pupil (exposed eye) points. If two or more points were labeled with a likelihood smaller than 0.1 (e.g., during blinks), we did not fit an ellipse but flagged the frame as missing data. We then applied the following signal processing to the pupil (exposed eye) time series of each measurement session: (i) resampling to 50 Hz; (ii) blinks were detected by a custom algorithm that marked outliers in the z-scored temporal derivative of the pupil size time series; (iii) linear interpolation of missing or poor data due to blinks (interpolation time window, from 150 ms before until 150 ms after missing data); (iv) low-pass filtering (third-order Butterworth, cut-off: 3 Hz); and (v) conversion to percentage of the 99.9 percentile of the time series.

*Quantification of VNS-evoked responses.* We quantified VNS-evoked pupil (or exposed eye) responses as the mean pupil size (exposed eye area) from 2.5 s to 7.5 s after VNS onset (colored bars in Fig. 2A), with the mean pre-trial baseline pupil size (exposed eye area) in the 5 s before VNS subtracted out. The background luminance during the calcium imaging was lower than for the other experiments and we observed substantially longer lasting pupil responses (Supplementary Fig. 6D). Therefore, for the calcium imaging sessions, we quantified VNS-evoked pupil responses as the mean pupil size from 10 s to 30 s after VNS onset (colored bars in Supplementary Fig. 6D), with the mean pre-trial baseline pupil (exposed eye) size in the 20 s before VNS subtracted out.

*Correction for reversion to the mean.* Pupil size (or exposed eye) also fluctuates spontaneously—that is, in the absence of vagus nerve stimulation. Importantly, when the pupil is relatively dilated it generally tends to constrict (and vice versa), a tendency that is called "reversion to the mean". Reversion to the mean predicts that baseline pupil size and evoked responses are anti-correlated, as is commonly observed[52,53]. Separately per type of experiment (see "Vagus nerve stimulation protocol"), we corrected the VNS-evoked pupil responses for a general (not related to VNS) reversion to the mean. To do so, we first characterized the spontaneous tendency of the pupil to change size on the time scale of VNS-evoked pupil dilation, as a function of the baseline pupil size. We collected the pupil time series from each 30 s window before VNS and sorted these into eight groups based on the mean pupil size in a pseudo-baseline window within this 30 s. Plotting the average pupil time series in each of these eight bins revealed a strong tendency for the pupil size to revert to the mean on the time scale of stimulation (Supplementary Fig. 3E, left). This same mean reversion was apparent in baseline-sorted VNS-locked pupil responses (Supplementary Fig. 3E, middle). A subtraction of pseudo- from VNS-locked pupil time series successfully isolated the phasic VNS-evoked dilation from mean reversion (Supplementary Fig. 3E, right). Next, we calculated a correction function from a fit of a cubic function to the spontaneous (pseudo-VNS) tendency to mean-revert in the same time windows used to measure VNS-evoked pupil dilation (Supplementary Fig. 3F; see black bars at bottom of Supplementary Fig. 3E). Finally, we predicted pupil (or exposed eye) response measures (reflecting

the predicted reversion to the mean) based on the observed pre-VNS baseline measures and subtracted those values from the observed VNS-evoked responses measures. All results are qualitatively the same without this correcting for a general reversion to the mean.

**Analysis of eye movement data**

*Preprocessing.* Eye position was computed as the center (*xy* coordinates) of the ellipse that was fitted to the detected pupil points (see "Analysis of pupil and eyelid data"). We then applied the following signal processing to the eye position time series (*x* and *y*) of each measurement session: (i) resampling to 50 Hz; (ii) blinks were detected by a custom algorithm that marked outliers in the z-scored temporal derivative of the pupil size time series; (iii) linear interpolation of missing or poor data due to blinks (interpolation time window, from 150 ms before until 150 ms after missing data); and (iv) conversion to z-scores.

*Quantification of VNS-evoked eye movements.* We used a Hanning window with a length of 1 s to compute a time-frequency representation decomposition of the epoched eye position time series (Supplementary Fig. 5A–C, G–I). Power in each frequency was expressed as the percent signal change with respect to the prestimulation baseline (baseline window, −5.5 to −0.5 s from VNS). In grounded animals, we observed that eye movements were phase-locked to the VNS train rate (red bands at 5, 10, and 20 Hz in Supplementary Fig. 5G–I). We quantified phase-locked VNS-evoked eye movements by taking the mean power during the 10 s VNS train at the frequency of stimulation (5, 10, or 20 Hz).

**Analysis of walking data**. The instantaneous velocity data was resampled to 50 Hz. We quantified VNS-evoked walking speed as the mean speed in the 10 s after VNS onset. We defined VNS-evoked walking probability as the fraction of pulses for which the absolute walking speed exceeded 0.05 cm/s (Supplementary Figs. 7A and 8A). We corrected for reversion to the mean of walking data in the same way as done for the VNS-evoked pupil responses (see above). All results are qualitatively the same without this correcting for a general reversion to the mean. In 18 of 45 ungrounded intact nerve parameter exploration sessions walking data was not collected (Fig. 5).

**Analysis of calcium imaging data**

*Preprocessing.* All frames in the image stack were co-registered using a custom two-step approach that we contributed to the Suite2p master repository on Github[61]. The steps were: (i) running Suite2p's default rigid registration step and computing a new reference image as the mean across the co-registered frames, and (ii) rerunning Suite2p's rigid registration using this new (enhanced) reference. We defined one region of interest per session as follows. First, for each pixel, we computed VNS-evoked calcium responses as the mean fluorescence from 2.5 to 7.5 s after VNS onset (colored bars in Fig. 4B), with the mean pre-trial baseline fluorescence in the 5 s before VNS subtracted out. Second, we tested all calcium response measures against 0 (irrespective of VNS parameters; Fig. 4A), using a cluster-based one-sample *t* test that corrected for multiple comparisons. Third, we defined the region of interest as all pixels whose time series exhibited significant VNS-evoked calcium responses ($P < 0.01$; cluster-corrected). Per session, we then averaged the fluorescence time series of pixels within the region of interest and applied: (i) resampling to 50 Hz, (ii) low-pass filtering (third-order Butterworth, cut-off: 3 Hz), (iii) linear de-trending of bleach-related rundown in the calcium signals, and (iv) rescaled to the percentage of the 99.9 percentile of the time series values.

*Quantification of VNS-evoked responses.* We quantified VNS-evoked calcium responses as the mean fluorescence from 2.5 to 7.5 s after VNS onset (colored bars in Fig. 4B), with the pre-trial baseline fluorescence in the 5 s before VNS subtracted out. We corrected for reversion to the mean in the same way as done for the VNS-evoked pupil responses (see above). All results are qualitatively the same without this correcting for a general reversion to the mean.

**Analysis of parametric VNS dependence**

*Log-logistic function.* The relationship between VNS parameters (amplitude, width, and rate) and VNS-evoked responses (pupil, calcium, walking, and exposed eye) was modeled with the following multivariate log-logistic function:

$$P = \frac{s}{1 + (A/a_1)^{-b_1}} \times \frac{1}{1 + (W/a_2)^{-b_2}} \times \frac{1}{1 + (R/a_3)^{-b_3}} \tag{3}$$

where $P$ was a vector of the VNS-evoked pupil (or calcium, walking, exposed eye) responses, $A$ was a vector of the pulse amplitudes, $W$ was a vector of the pulse widths, $R$ was a vector of the pulse rates, and $s$, $a_1$, $a_2$, $a_3$, $b_1$, $b_2$, $b_3$ were the free parameters of the fit. Likewise, for the reduced model, the relationship between VNS parameters (charge/pulse and rate) and VNS-evoked responses (pupil, calcium, walking, exposed eye) was modeled with:

$$P = \frac{s}{1 + (C/a_1)^{-b_1}} \times \frac{1}{1 + (R/a_2)^{-b_2}} \tag{4}$$

where **C** was a vector the pulse charges (amplitude × width) and the other parameters are the same as in Eq. (3).

*Nerve engagement*. We constructed vector *N* of the predicted VNS-evoked nerve engagement in a way that met two objectives: (i) reducing the three-dimensional VNS parameter space (amplitude × width × rate) to a one-dimensional independent variable, and (ii) allowing all interactions to be modeled with linear regressions (instead of the log-logistic functions). We constructed vector *N* with a 20-fold cross-validation procedure. In every iteration, 95% of the data were used to fit the relationship between VNS parameters (amplitude, width, and rate) and VNS-evoked pupil responses with the log-logistic function described above; the remaining 5% of the data were used to predict VNS-evoked nerve engagement by plugging the corresponding pulse amplitudes, widths and rates in the fitted function. As expected, nerve engagement was linearly correlated with VNS-evoked pupil and calcium responses (Supplementary Fig. 8B).

*Mediation analysis*. We performed a mediation analysis to characterize the interaction between VNS parameters, VNS-evoked pupil responses, and VNS-evoked calcium responses (Fig. 6). We fitted the following linear regression models based on standard mediation path analysis:

$$\mathbf{P} = i_0 1 + c\mathbf{N} \tag{5}$$

$$\mathbf{C} = i_1 1 + a\mathbf{N} \tag{6}$$

$$\mathbf{P} = i_2 1 + c'\mathbf{P} + b\mathbf{C} \tag{7}$$

where **P** was a vector of VNS-evoked pupil responses, **C** was a vector of the VNS-evoked calcium responses, **N** was a vector of the predicted VNS-evoked nerve engagement due (see above), and $c$, $c'$, $a$, $b$, $i_0$, $i_1$, and $i_2$ were the free parameters of the fit. The parameters were fit using freely available R-software[62].

**Artifact rejection**. In the "parameter exploration" and "baseline pupil dependence" experiments, we excluded VNS events from the analyses for which we could not reliably record pupil size (due to blinking or other reasons; 1.51% and 1.92% of events, respectively). In the "axon imaging experiment", we excluded VNS events from the analyses for which we could not reliably record pupil size and/or observed more than 2 μm motion in either *x* or *y* direction (16.67% of events; Supplementary Fig. 6A).

**Statistical comparisons**. We used the one-sample *t* test to test for significant differences between VNS-evoked pupil, walking, or exposed eye responses and 0. We refrained from testing the VNS-evoked calcium responses against 0, which would be "double dipping" because of the way we defined our region of interest (see "Analysis of calcium imaging data"). Instead, we used the paired-sample *t* test to test for significant differences between VNS-evoked calcium responses in the highest four charge bins and those responses in the lowest charge bin. Thus, this test does not quantify the effect of VNS on evoked calcium responses per se but addresses the separate question of the parameter dependence of VNS-evoked calcium responses. In all cases, we corrected for multiple comparisons with false discovery rate (FDR; Benjamini–Hochberg procedure).

We used sequential polynomial regression analysis[63], to quantify whether the relationships of interest were better described by a first-order (linear) or a second-order (quadratic) model:

$$\mathbf{Y} = \beta_0 1 + \beta_1 \mathbf{X} + \beta_2 \mathbf{X}^2 \tag{8}$$

where **Y** was a vector of the dependent variable (e.g., VNS-evoked pupil responses), **X** was a vector of the independent variable (e.g., prestimulation baseline pupil size measures), and $\beta$ as polynomial coefficients. To assess the amount of variance that each predictor accounted for independently, we orthogonalized the regressors prior to model fitting using QR-decomposition. Starting with the zero-order (constant) model and based on F-statistics[63], we tested whether incrementally adding higher-order predictors improves the model significantly (explains significantly more variance). We tested zero-order up to second-order models.

Using the Bayesian information criterion (BIC), we additionally checked whether the added complexity of a quadratic model was justified to account for the data. Formally comparing models in this way gave identical results. A difference in BIC of 10 is generally taken as a threshold for considering one model a sufficiently better fit. The dependence of VNS-evoked pupil response on baseline pupil size (Supplementary Fig. 3G) was better described by a quadratic model (BIC = 6721) than a linear model (BIC = 6736). All other cases (Fig. 6 and Supplementary Fig. 8) were better described by linear models.

We used the non-parametric cluster-level paired *t* test within the MNE implementation[64,65] to test pixelwise VNS-evoked calcium responses against 0 (Fig. 4A). The algorithm implemented 10 K randomly generated permutations of the data to perform a Monte Carlo-style permutation test. This procedure was robust with respect to inflated false-positive rates. We used a cluster-correction threshold of $P < 0.01$.

All tests were performed two-tailed.

**Reporting summary**. Further information on research design is available in the Nature Research Reporting Summary linked to this article.

## Data availability
All time-series data and the pre-trained DeepLabCut[35] network are publicly available on https://doi.org/10.6084/m9.figshare.12899375. Source data are provided with this paper.

## Code availability
Analysis scripts[66] are publicly available on https://doi.org/10.5281/zenodo.4243062.

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

## Acknowledgements

We thank Anton Banta for technical assistance, Hannah Ramsaywak and Marisa Hudson for help with animal care and handling, Sarim Aleem for help with pupil size analysis, Michael Kilgard's lab for training in general VNS techniques, and Kip Ludwig, David McCormick, and Robert Froemke for helpful discussions.

## Author contributions

Z.M.: investigation, methodology, and writing—review and editing; J.W.d.G.: formal analysis, methodology, software, writing—original draft, and writing—review and editing; Y.S.: investigation and data curation; R.A.: conceptualization and funding acquisition; A.S.: formal analysis and supervision; M.P.W.: methodology, software, and resources; W.Z.: methodology and software; M.J.M.: conceptualization, investigation, formal analysis, writing—original draft, writing—review and editing, funding acquisition, supervision, and project administration.

## Competing interests

The authors declare no competing interests.
