## [Peer Review File · Nature Communications]

Reviewer #1 (Remarks to the Author):

Mridha et al. investigated the effects of vagus nerve stimulation (VNS) on pupil dilation, locomotion and on the activity of cholinergic axons in the auditory cortex. They found that VNS stimulation reliably leads to pupil dilation and cholinergic activity in an optimally chosen stimulation parameter regime. The authors argue that pupil dilation can therefore be used as a biosensor feedback for VNS stimulation. This is useful for therapeutic approaches that attempt to establish positive health effects of VNS in a range of CNS diseases.

Arguably the main benefit neuroscience could offer to VNS therapy would be a mechanistic understanding of VNS effects on the brain, based on which this branch of therapy could be refined and improved; every step towards this goal is useful. I thus find the present study of high importance. My main critique is that while the authors carefully optimize the experimental conditions and stimulation parameters, the analyses underlying the major findings is somewhat superficial and leaves the reader uncertain about the conclusions. Provided these uncertainties are lifted, this study might become an important milestone in VNS research. My specific comments are below.

Major points.

1. For pupil-dilation, the parameter space was thoroughly explored and data analysis was carried out laudably thoughtfully. At the same time, the cholinergic imaging is an important part of the study (the only one that provides some insights into the underlying brain mechanisms), where the same depth of analysis would be expected. In particular, was cholinergic response best explained as a function of charge, similar to pupil response (parallel analysis with that presented in Fig.3)? Did cholinergic response depend on pupil-indexed brain state?

2. The analysis of movement also seem to carry more potential and could be better analyzed. Was there also spontaneous movement, and if yes, how did it compare with VNS-evoked movement? Was absence of VNS-evoked movement observed in the transection experiments, ruling out an off-target origin? Did pre-stimulus movement affect pupil dilation (would be expected based on previous papers of the last Author)? Did it affect cholinergic response?

3. While I realize this is a subjective judgement, I find the mediation analysis (Fig.7) maybe the most exciting part of the study. However, the analysis is not fully convincing. 'We found that axonal response (Fig. 7A) and pupil dilation (Fig. 7B) exhibited a similar parameter dependence to that observed previously (without excluding walking trials).' – a more direct comparison of axonal response and pupil response would be more convincing. What was the pupil response like in ACh Ca-response bins (to have an idea on the transfer function)? 'VNS-evoked pupil dilation and BF-ACh axonal activity were robustly correlated (Fig. 7C).' – this perceived robustness is subjective; an R-square of $0.62^2=0.38$ shows that the majority of the variance remained unexplained. The conclusion of the mediation analysis (based on linear regression) could be aided by partial correlation calculations (which are rather simple). Even better, moving away from these linear techniques towards information theory-based measures that capture non-linear relationships as well would be even nicer (although less simple).

4. The Kilgard-lab has performed an excellent line of work, demonstrating the necessity of multiple major neuromodulatory systems (acetylcholine, noradrenaline, serotonin) for VNS-induced cortical plasticity. The involvement of the locus coeruleus is particularly well-studied. While the Authors do make some of these important references, the manuscript in the present form puts an undue emphasis on the cholinergic system. While I accept that a comparative approach involving the noradrenergic system is a major undertaking very likely beyond the scope of this study, it should be made clearer that acetylcholine is probably only one, though important, piece of the puzzle and other neuromodulators may play comparably important roles (for instance in mediating pupil dilation studied here).

Minor points

1. Abstract, first sentence – unclear what 'global' exactly refers to here

2. p.4 – ‘the multivariate log-logistic function fit well’. What do the Authors mean by ‘fit well’? A quantitative approach to illustrate this would be more elegant and convincing.
3. Stimulation rate seem to have had no effect on cholinergic response. Is there a clear explanation for this? Is this consistent with the charge-dependent explanation of the effects?
4. Fig.5 – panel label ‘E’ is missing
5. p. 12 ‘t-VNS though its auricular branch’ – through
6. Methods – ‘sample sizes’ section is confusing, a table could help
7. ‘For the ‘light control experiment’, we performed a total of two pupillometry sessions in one unique mice. – how was that mouse (use singular) ‘unique’?
8. p. 15 – ‘pre-exciting head post’ – revise
9. ‘We corrected for reversion to the mean of walking data in the same way as done for the VNS-evoked pupil responses’ – this was unclear to me. How exactly, and why was this correction for the walking data necessary?
10. Mediation analysis – I don’t understand how the vector ‘N’ was constructed. ‘data as used’ is probably a typo. Probably related to this, what is ‘nerve engagement (a.u.)’ in Fig.S7?
11. Statistics – serial hierarchical testing should be better explained; how were models with different order (hence different number of parameters) compared?
12. Statistics – last paragraph is impenetrable; a better explanation should be added.
13. p. 26 – each, not ‘eech’
14. p. 6 – ‘The charge at halfmaximal pupil response on double cut nerve ungrounded sessions (0.65 +/- 0.13 μC) was was significantly larger than the on double cut nerve grounded sessions’ – remove the typos from the sentence

Reviewer #2 (Remarks to the Author):

The authors report work showing that vagus nerve stimulation (VNS) promotes acetylcholine release in the neocortex, the efficacy of which can be tracked by pupil dilation. Parametrically varying the amplitude, pulse width, and train rate across 60 possible combinations, the authors identify a sweet spot (or “goldilocks” area) within the parameter space that maximally promotes pupil dilation without being subject to confounding current spread to fibers other than the vagus nerve. The exploration of optimal parameter settings is then refined and translated to acetylcholine release in the auditory cortex, again identifying a consistent goldilocks area in the parameter space. In addition, mediation analysis suggests pupil dilation is an excellent readout for inferring VNS-elicited acetylcholine release. Also of note is that the authors use a novel method of correcting for regression toward the mean in pupil-linked arousal studies that I suspect will become gold standard for studies moving forward. The work is timely, of crucial importance to the field, of broad interest beyond those interested in VNS, and has been implemented and written up with exceptional skill. I have only few points to raise.

1. Though pupil dilation has been linked to both acetylcholine (ACh) and norepinephrine (NE), the authors do not spend enough space dealing with this fact and some of the confusion that could arise related to this. As the authors, reviewers and editor(s) well know, NE is the excitatory neurotransmitter of the sympathetic nervous system, and would be most directly responsible for a dilation. Whereas ACh is the excitatory neurotransmitter of the parasympathetic nervous system which would be directly responsible for a pupil constriction. I understand the ACh release in the

neocortex is caused by VNS, and that pupil dilation is only a nice readout of the efficacy of VNS for doing just that, but I think this needs to be spelled out and the potential confusion addressed directly. That is, the paper as it is can easily be mistakenly read to suggest VNS causes ACh release which causes the pupil to dilate. The mediation analysis could also promote confusion as it again could easily be mistakenly interpreted to mean that VNS causes pupil dilation by promoting ACh release. The authors have seemingly overlooked this potential source of confusion simply because the true meaning of the work is not that VNS produces ACh activity in the parasympathetic nervous system, but walking the potentially confused reader through why it is not would make this paper more accessible.

There are multiple places in the paper where clarifying specific interest in ACh though NE is playing a role as well, acknowledging the direct path causing the pupil to dilate (excitation of SNS or inhibition of PNS or both), or just specifying the ACh is not directly causing pupil dilation is warranted, eg:

"However, it has not been demonstrated whether VNS actually evokes acetylcholine release from basal forebrain into the cortex, so the role of acetylcholine in VNS-evoked cortical plasticity remains speculative."

"We further show that this VNS-evoked dilation is mediated by a network involving release of acetylcholine from basal forebrain into neocortex and interacts non-linearly with the current momentary state of the brain."

"We observed that VNS trains elicited pupil dilation (Fig. 1E), cholinergic axon activity (Fig. 1F), and probabilistically evoked walking on the treadmill (Fig. 1G). These evoked responses occurred against the backdrop of the ongoing dynamics of these signals across each session (Fig. 1H)."

"The eye lid receives the same sympathetic innervation as the pupil dilator, but no parasympathetic input (Loewenfeld and Lowenstein, 1993). Therefore, we measured the area of exposed eye. Eye lid exhibited a similar pattern of parameter dependence to pupil dilation."

"[...] suggesting a prominent sympathetic contribution to the VNS-evoked pupil response (Fig. S2B-D)."

"We previously showed that long-lasting (e.g. >2-3 seconds) pupil dilation closely tracks cholinergic axonal activity in auditory and visual cortex in similar behavioral conditions to those used in this study (Reimer et al., 2016). Therefore, we suspected that VNS-evoked pupil dilation was linked to release of acetylcholine from basal forebrain into the cortex (BF-ACh)."

"We found that the indirect path of VNS driving ACh, in turn driving pupil dilation, statistically mediates the majority of the apparent effect of VNS on pupil (Fig. 7E)."

NOTE: I do not think it is necessary to rewrite all these excerpt to address my comment – a couple of sentences in the introduction and a short paragraph in the discussion would probably suffice.

2. I understand that the "goldilocks" zone is the parameter space that is "just right" because it produces pupil dilation when the nerve is intact, and does not when the nerve is cut. Meaning that these parameter settings will allow pupil dilation to serve as a good readout of VNS efficacy, whereas lower settings do not provoke pupil dilation, and higher settings provoke pupil dilation through current spread as well as from activation through the vagus nerve (or even without activation through the vagus nerve when it has been cut). I like the term for the personality it shows. However, it brings to mind a Japanese visitor to my lab in grad student, who made a heartfelt plea during his talk for scientists with a good command of the English language to refrain from stylistic turns and instead focus on clarity. This man had clearly struggled many times with reading in English and I suspect the Goldilocks reference would be one of those times that when he finally was able to piece together what it meant, he would be frustrated beyond belief that it was not given a label that was direct and clear to non-native English speakers.

3. Also related to the goldilocks zone, I think that figures 3D and 3F could be reorganized or in some way combined to emphasize that the goldilocks zone is defined by effects in both the intact and double cut nerve data.

4. This sentence was interesting to me and I'd like a little more explanation and/or speculation: "When the animal was grounded, substantial dilation occurred during VNS with intact, single or double cut nerve (Fig. S3G-L), as well as substantial offset dilation subsequent to the end of each VNS train (Fig. S3G-L), which was not observed in ungrounded conditions (Fig. S3A-F)."

5. I'm embarrassed to ask this, but I'm never used mediation analysis before, and am only

moderately familiar with how it works. How is the directionality determined? That is, how does the model of VNS driving ACh driving pupil dilation account for more variance than a model in which VNS drives pupil dilation which drives ACh?

Thank you for thoughtful comments and for your overall positive assessment of our work. Please see below our detailed responses (in black) to the reviewer comments (in blue). At the reviewer's request, we have added many additional analyses to clarify and substantiate our conclusions. In addition, we have improved our deep net for pupil size extraction, resulting in a substantial increase (a doubling) in explained variance in our main "parameter exploration" experiment. We believe that these revisions have substantially improved our manuscript by strengthening the support for our original conclusions, which are unchanged. We hope that you will now find our manuscript suitable for publication in *Nature Communications*.

Sincerely, on behalf of all authors,
Jan Willem de Gee, Zakir Mridha, and Matthew McGinley

Reviewer #1 (Remarks to the Author):

Mridha et al. investigated the effects of vagus nerve stimulation (VNS) on pupil dilation, locomotion and on the activity of cholinergic axons in the auditory cortex. They found that VNS stimulation reliably leads to pupil dilation and cholinergic activity in an optimally chosen stimulation parameter regime. The authors argue that pupil dilation can therefore be used as a biosensor feedback for VNS stimulation. This is useful for therapeutic approaches that attempt to establish positive health effects of VNS in a range of CNS diseases.

Arguably the main benefit neuroscience could offer to VNS therapy would be a mechanistic understanding of VNS effects on the brain, based on which this branch of therapy could be refined and improved; every step towards this goal is useful. I thus find the present study of high importance. My main critique is that while the authors carefully optimize the experimental conditions and stimulation parameters, the analyses underlying the major findings is somewhat superficial and leaves the reader uncertain about the conclusions. Provided these uncertainties are lifted, this study might become an important milestone in VNS research. My specific comments are below.

We are pleased that you consider our mechanistic work of 'high importance' and useful in guiding future therapeutic approaches. Please find our replies to your specific comments below.

Major points.

1. For pupil-dilation, the parameter space was thoroughly explored and data analysis was carried out laudably thoughtfully. At the same time, the cholinergic imaging is an important part of the study (the only one that provides some insights into the underlying brain mechanisms), where the same depth of analysis would be expected. In particular, was cholinergic response best explained as a function of charge, similar to pupil response (parallel analysis with that presented in Fig.3)?

We are pleased with the reviewer's request, agree that these additional analyses are valuable, and now include them. Specifically, we now include analyses in Fig. 4 on VNS-evoked calcium responses that parallel those in Fig. 3 on VNS-evoked calcium responses. We added calcium response separately for each pulse width, as a function of charge/pulse (Fig. 4E). For the VNS-evoked pupil responses, the curves for different pulse widths lie approximately on top of each other (Fig. 3A), supporting this parameter reduction strategy. We have also added VNS-evoked calcium timeseries split by charge bin (Fig. S5H). Finally, we have now included that modeling

the VNS-evoked calcium response as a function of pulse amplitude, pulse width and train rate explained 49.11% of the variance, and as function of pulse charge and train rate explained 48.89% (Fig. 4F). This further supports our parameter reduction strategy.

Did cholinergic response depend on pupil-indexed brain state?

Although we agree this is an important question, the “axon imaging experiment” explored a VNS parameter space, and thus was not designed for, or well-suited for, addressing this question.

We were able to determine if phasic pupil responses dependent on baseline (pre-stimulation) pupil size, because in the “baseline pupil experiment” we repeated a single parameter combination many times in each session (see main text). In contrast, the axon imaging and pupil parameter exploration experiments involved parametric VNS, and thus VNS-evoked calcium and pupil responses depended strongly on VNS parameters (pulse amplitude, pulse width and train rate) on each trial. These experiments were therefore not likely to be sensitive to the effects of baseline pupil-indexed brain state on evoked calcium or pupil responses.

To nonetheless attempt this analysis, we first regressed out the variation in VNS-evoked calcium responses (Fig. R1A, left) and pupil responses (Fig. R1B, left) that was predicted from the charge/pulse, and then asked whether the residuals depended on pupil-indexed brain state. There was no effect for the calcium responses (Fig. R1A, right). Additionally, VNS-evoked pupil responses in the pupil parameter exploration experiments also did not strongly depend on pupil-indexed brain state, and only weakly showed a hint of the inverted-U dependence on baseline (Fig. R1B, right).

Upon closer inspection, we found that the baseline pupil dependence of VNS-evoked pupil responses was not particularly strong even in “baseline pupil experiment”. As a result, we have now moved these results to a supplement (Fig. S2E-G).

Fig. R1. (A) Left: VNS-evoked calcium response measures separately per charge/pulse bin and train rate. Colored lines, fitted log logistic function (Materials and Methods); error bars, s.e.m. \times 1.96 (95% confidence interval of the mean). Middle, as left, after regressing out dependence on charge. Right: scatterplot of the relationship between residual VNS-evoked calcium responses and pre-stimulation baseline pupil size. Data points are individual VNS trains; red line, linear fit; Pearson correlation, $p = 0.14$. A 1st-order (linear) fit was not superior to a constant fit ($F = 2.21$, $p = 0.14$) and a 2nd-order (quadratic) fit was not superior to the first-order fit ($F = 1.51$, $p = 0.22$) (sequential polynomial regression; Materials and Methods). There was no evidence for a non-monotonic (inverted-U) relationship between residual VNS-evoked calcium responses and pre-stimulation baseline pupil size (tested via sequential polynomial regression; Materials and Methods). **(B)** As panel A, but for VNS-evoked pupil responses in the “parameter exploration experiment”. Pearson correlation, $p < 0.001$. A 1st-order (linear) fit was superior to a constant fit ($F = 62.06$, $p < 0.001$) and a 2nd-order (quadratic) fit was not superior to the first-order fit ($F = 2.87$, $p = 0.09$).

2. The analysis of movement also seem to carry more potential and could be better analyzed. Was there also spontaneous movement, and if yes, how did it compare with VNS-evoked movement?

Indeed, there were spontaneous movements (which we corrected for in the main Fig. 5; see the Materials and Methods). We now explicitly present the spontaneous movements, and the relationship to evoked movements, in new panels Fig. S6B-E.

Was absence of VNS-evoked movement observed in the transection experiments, ruling out an off-target origin?

We did not observe VNS-evoked changes in movement in the double cut experiments in all but the highest charge bin, which we now present in panels Fig. S6F-I. These results mimic those based on VNS-evoked pupil responses (Fig. 3F), suggesting that indeed there is no off-target driving of locomotor activity or pupil dilation, except for those very large charges which we consistently show have off target effects with intact nerve, as well.

Did pre-stimulus movement affect pupil dilation (would be expected based on previous papers of the last Author)? Did it affect cholinergic response?

For the same reasons as above (see response to ‘Did cholinergic response depend on pupil-indexed brain state?’) the imaging data set was not effective for determining how cholinergic responses depended on brain state parameters, because they explored a stimulation parameter space and thus had large parameter-related ‘variability’ in responses that would swamp state-dependent effects.

As above, we attempted this analysis by regressing out the parameter dependence of responses. As expected from our previous work, baseline walking speed was linearly correlated to baseline calcium (Fig. R2A) and pupil size (Fig. R2B). However, the residuals of evoked calcium (Fig. R2C) and pupil (Fig. R2D) showed no (or a very weak) dependence on baseline speed. As above, we do not interpret that is a true ‘negative result’ because the experiment was not well matched to the question. Therefore, we do not think it is appropriate to include these results in the manuscript.

Fig. R2. Dependence of baseline and evoked calcium and pupil on baseline running speed. (A) Scatterplot of the relationship between baseline (pre-stimulation) calcium activity and walking speed. Data points are individual VNS trains; red line, linear fit; Pearson correlation, $p < 0.001$. A 1st-order fit was superior to a constant fit ($F = 51.14$, $p < 0.001$) and a 2nd-order fit was not superior to the first-order fit ($F = 0.04$, $p = 0.85$) (sequential polynomial regression; see Materials and Methods). (B) As A, but for the relationship between baseline (pre-stimulation) pupil size and walking speed. Pearson correlation, $p < 0.001$. A 1st-order fit was superior to a constant fit ($F = 41.75$, $p < 0.001$) and a 2nd-order fit was not superior to the first-order fit ($F = 1.74$, $p = 0.19$). (C) As A, but for the relationship between VNS-evoked calcium responses and baseline (pre-stimulation) walking speed. Pearson correlation, $p = 0.84$. A 1st-order fit was not superior to a constant fit ($F = 0.04$, $p = 0.83$) and a 2nd-order fit was not superior to the first-order fit ($F = 0.23$, $p = 0.63$). (D) As A, but for the relationship between VNS-evoked pupil responses and baseline (pre-stimulation) walking speed. Pearson correlation, $p = 0.001$. A 1st-order fit was superior to a constant fit ($F = 12.06$, $p = 0.001$) and a 2nd-order fit was not superior to the first-order fit ($F = 0.76$, $p = 0.38$).

3. While I realize this is a subjective judgement, I find the mediation analysis (Fig.7) maybe the most exciting part of the study. However, the analysis is not fully convincing. 'We found that axonal response (Fig. 7A) and pupil dilation (Fig. 7B) exhibited a similar parameter dependence to that observed previously (without excluding walking trials).' – a more direct comparison of axonal response and pupil response would be more convincing. What was the pupil response like in ACh Ca-response bins (to have an idea on the transfer function)?

We are glad the reviewer found the mediation analysis exciting! To make the analysis more convincing, and show the transfer function, we have now added binned data to the scatter plots in Fig. 6 & Fig. S7, as requested. We have also added partial correlations, which show that pupil and cholinergic responses are correlated even after regressing out their dependence on VNS parameters (new Fig. 6D & Fig. S7E). To check for non-linear effects, we tested whether a first or second order polynomial model provided a better description of this relationships (using sequential polynomial regression; see Materials and Methods). In all cases, a first order (linear) relationship provided the best fit. We report the F and p values and information criterion for each of the polynomial regressions, throughout the manuscript.

'VNS-evoked pupil dilation and BF-ACh axonal activity were robustly correlated (Fig. 7C).' – this perceived robustness is subjective; an R-square of $0.62^2=0.38$ shows that the majority of the variance remained unexplained.

We agree that “robustly” is subjective, and now have removed this word from the sentence. We also agree that the majority of variance remains unexplained. We believe the reason for this is three-fold. First, given that we are looking at the resolution of individual axons and comparing to an indirect readout of brain state (pupil size), we feel even 38% of variance is actually quite high, although we accept that this is subjective. Second, measurement noise in both VNS-evoked pupil and ACh responses will decrease the maximum expected explained variance. Third, recent insights indicate that pupil responses not only track responses of the basal forebrain cholinergic system, but also responses of the noradrenergic (NE) locus coeruleus (de Gee et al., 2017; Joshi et al., 2016; Reimer et al., 2016), the serotonergic dorsal raphe (Cazettes et al., 2020) and perhaps even the dopaminergic (DA) midbrain (de Gee et al., 2017). We thus expect that adding NE, 5HT and/or DA responses as predictors would substantially increase the explained variance. However, the simultaneous imaging of various neuromodulatory axons was outside the scope of this project. We have now added a Discussion paragraph providing this context (p. 14); please also see our replies to Reviewer 1 major comment #4, and to Reviewer 2 major comment #1 for additional relevant comments.

The conclusion of the mediation analysis (based on linear regression) could be aided by partial correlation calculations (which are rather simple). Even better, moving away from these linear techniques towards information theory-based measures that capture non-linear relationships as well would be even nicer (although less simple).

We agree it is useful to apply multiple analyses approaches, and now have also added partial correlations, which show that pupil and cholinergic responses are correlated even after regressing out their dependence on VNS parameters (Fig. 6D & Fig. S7C,E). To address non-linearity, as mentioned above, we now tested whether a first or second order polynomial model provided a better description of this relationships (with sequential polynomial regression; see Materials and Methods). In all cases, a first order (linear) relationship provided the best fit; we report these results in the text. Given that the sequential polynomial regression consistently pointed to linear relationships, we expect that an information theory-based measure would not yield additional insight (or much increase in explained variance).

4. The Kilgard-lab has performed an excellent line of work, demonstrating the necessity of multiple major neuromodulatory systems (acetylcholine, noradrenaline, serotonin) for VNS-induced cortical plasticity. The involvement of the locus coeruleus is particularly well-studied. While the Authors do make some of these important references, the manuscript in the present form puts an undue emphasis on the cholinergic system. While I accept that a comparative approach involving the noradrenergic system is a major undertaking very likely beyond the scope of this study, it should be made clearer that acetylcholine is probably only one, though important, piece of the puzzle and other neuromodulators may play comparably important roles (for instance in mediating pupil dilation studied here).

We agree that multiple neuromodulators are involved in VNS-evoked brain state changes. We apologize for not adequately discussing this in the 1st submission, and now do so more extensively (p. 1-2 & 14). We also agree that imaging noradrenergic axons was beyond the scope of this project. We have added a discussion paragraph on the roles of other modulators, which we believe will strengthen the paper by providing a better context. We thank the reviewer for pushing us to do this.

Minor points

1. Abstract, first sentence – unclear what ‘global’ exactly refers to here

We changed ‘global’ to ‘brain-wide release of’.

2. p.4 – ‘the multivariate log-logistic function fit well’. What do the Authors mean by ‘fit well’? A quantitative approach to illustrate this would be more elegant and convincing.

We now report R^2 (% explained variance) in the figures, and associated p-values in the text, throughout.

3. Stimulation rate seem to have had no effect on cholinergic response. Is there a clear explanation for this? Is this consistent with the charge-dependent explanation of the effects?

We agree that this is somewhat surprising. The effect is consistent with the charge-dependent explanation, because all of that analysis is based on charge/pulse, which should quite directly (albeit perhaps non-linearly) relate to # of recruited nerve fibers. Our interpretation is that short-term synaptic plasticity, and/or spike rate adaptation, and/or network-level adaptation, at any stage(s) between the vagus nerve and cholinergic basal forebrain, could account for this. Because this is so highly speculative, we chose not to (over)interpret it.

4. Fig.5 – panel label ‘E’ is missing

Fixed.

5. p. 12 ‘t-VNS though its auricular branch’ – through

Fixed.

6. Methods – ‘sample sizes’ section is confusing, a table could help

Done.

7. ‘For the ‘light control experiment’, we performed a total of two pupillometry sessions in one unique mice. – how was that mouse (use singular) ‘unique’?

Fixed.

8. p. 15 – ‘pre-exciting head post’ – revise

Fixed.

9. ‘We corrected for reversion to the mean of walking data in the same way as done for the VNS-evoked pupil responses’ – this was unclear to me. How exactly, and why was this correction for the walking data necessary?

We added a new paragraph in the Materials and Methods, and new panels Fig. S6B-E to clarify this. The correction was not ‘necessary,’ in that the conclusions are the same without this correction. However, we feel this correction makes the results more cleanly interpretable (albeit

more complicated). As mentioned by reviewer #2, we think this mean-reversion correction will be helpful for the pupillometry field, in general. We have also expanded the Materials and Methods to explain this approach in more depth (p. 20).

10. Mediation analysis – I don't understand how the vector 'N' was constructed. 'data as used' is probably a typo. Probably related to this, what is 'nerve engagement (a.u.)' in Fig.S7?

'Data as used' should have been 'data was used'. We now created a separate section in the Materials and Methods and outline how (and why) we constructed 'nerve engagement' (p. 22).

11. Statistics – serial hierarchical testing should be better explained; how were models with different order (hence different number of parameters) compared?

The models were compared via F-tests, which test whether a higher order model explains *significantly* more variance. We now explain this more clearly (p. 23). Additionally, we have added model comparison based on Bayesian information criterion (BIC; Materials and Methods; p. 23), which is a complementary approach to test whether a more complex model is justified to account for the data. Formally comparing models in this way gave identical results and we report both approaches.

12. Statistics – last paragraph is impenetrable; a better explanation should be added.

We have expanded this section (p. 23-24).

13. p. 26 – each, not 'eech'

Fixed.

14. p. 6 – 'The charge at halfmaximal pupil response on double cut nerve ungrounded sessions (0.65 +/- 0.13 μC) was was significantly larger than the on double cut nerve grounded sessions' – remove the typos from the sentence

Fixed.

Reviewer #2 (Remarks to the Author):

The authors report work showing that vagus nerve stimulation (VNS) promotes acetylcholine release in the neocortex, the efficacy of which can be tracked by pupil dilation. Parametrically varying the amplitude, pulse width, and train rate across 60 possible combinations, the authors identify a sweet spot (or "goldilocks" area) within the parameter space that maximally promotes pupil dilation without being subject to confounding current spread to fibers other than the vagus nerve. The exploration of optimal parameter settings is then refined and translated to acetylcholine release in the auditory cortex, again identifying a consistent goldilocks area in the parameter space. In addition, mediation analysis suggests pupil dilation is an excellent readout for inferring VNS-elicited acetylcholine release. Also of note is that the authors use a novel method of correcting for regression toward the mean in pupil-linked arousal studies that I suspect will become gold standard for studies moving forward. The work is timely, of crucial importance to the field, of broad interest beyond those interested in VNS, and has been implemented and written up with exceptional skill. I have only few points to raise.

We thank the reviewer for their kind words and high appreciation for our work.

1. Though pupil dilation has been linked to both acetylcholine (ACh) and norepinephrine (NE), the authors do not spend enough space dealing with this fact and some of the confusion that could arise related to this. As the authors, reviewers and editor(s) well know, NE is the excitatory neurotransmitter of the sympathetic nervous system, and would be most directly responsible for a dilation. Whereas ACh is the excitatory neurotransmitter of the parasympathetic nervous system which would be directly responsible for a pupil constriction. I understand the ACh release in the neocortex is caused by VNS, and that pupil dilation is only a nice readout of the efficacy of VNS for doing just that, but I think this needs to be spelled out and the potential confusion addressed directly. That is, the paper as it is can easily be mistakenly read to suggest VNS causes ACh release which causes the pupil to dilate. The mediation analysis could also promote confusion as it again could easily be mistakenly interpreted to mean that VNS causes pupil dilation by promoting ACh release. The authors have seemingly overlooked this potential source of confusion simply because the true meaning of the work is not that VNS produces ACh activity in the parasympathetic nervous system, but walking the potentially confused reader through why it is not would make this paper more accessible.

We agree, and apologize for omitting the important role of VNS driven NE release, and for not more extensively clarifying that basal forebrain acetylcholine release is NOT the same as acetylcholine release in the parasympathetic nervous system (including at the level of the pupil muscles). We have added a discussion paragraph to make this clearer (p. 14).

There are multiple places in the paper where clarifying specific interest in ACh though NE is playing a role as well, acknowledging the direct path causing the pupil to dilate (excitation of SNS or inhibition of PNS or both), or just specifying the ACh is not directly causing pupil dilation is warranted, eg:

“However, it has not been demonstrated whether VNS actually evokes acetylcholine release from basal forebrain into the cortex, so the role of acetylcholine in VNS-evoked cortical plasticity remains speculative.”

To make this clearer, we have added “so the role of cortical acetylcholine in VNS-evoked cortical plasticity remains speculative.”

“We further show that this VNS-evoked dilation is mediated by a network involving release of acetylcholine from basal forebrain into neocortex and interacts non-linearly with the current momentary state of the brain.”

We have added a sentence here: “This network very likely involves other neuromodulators, as well, such as norepinephrine.”

“We observed that VNS trains elicited pupil dilation (Fig. 1E), cholinergic axon activity (Fig. 1F), and probabilistically evoked walking on the treadmill (Fig. 1G). These evoked responses occurred against the backdrop of the ongoing dynamics of these signals across each session (Fig. 1H).”

We have modified to: “cholinergic axon activity in auditory cortex”

“The eye lid receives the same sympathetic innervation as the pupil dilator, but no parasympathetic input (Loewenfeld and Lowenstein, 1993). Therefore, we measured the area of exposed eye. Eye lid exhibited a similar pattern of parameter dependence to pupil dilation.”

“[...] suggesting a prominent sympathetic contribution to the VNS-evoked pupil response (Fig. S2B-D).”

We think this is clear, as is, given the other changes we are making.

“We previously showed that long-lasting (e.g. >2-3 seconds) pupil dilation closely tracks cholinergic axonal activity in auditory and visual cortex in similar behavioral conditions to those used in this study (Reimer et al., 2016). Therefore, we suspected that VNS-evoked pupil dilation was linked to release of acetylcholine from basal forebrain into the cortex (BF-ACh).”

We think this is clear, as is, given the other changes we are making.

“We found that the indirect path of VNS driving ACh, in turn driving pupil dilation, statistically mediates the majority of the apparent effect of VNS on pupil (Fig. 7E).”
NOTE: I do not think it is necessary to rewrite all these excerpt to address my comment – a couple of sentences in the introduction and a short paragraph in the discussion would probably suffice.

As mentioned above, we have added a paragraph to the Discussion (p. 14), and we have also added clarifying text to the Introduction (p. 1-2).

2. I understand that the “goldilocks” zone is the parameter space that is “just right” because it produces pupil dilation when the nerve is intact, and does not when the nerve is cut. Meaning that these parameter settings will allow pupil dilation to serve as a good readout of VNS efficacy, whereas lower settings do not provoke pupil dilation, and higher settings provoke pupil dilation through current spread as well as from activation through the vagus nerve (or even without activation through the vagus nerve when it has been cut). I like the term for the personality it shows. However, it brings to mind a Japanese visitor to my lab in grad student, who made a heartfelt plea during his talk for scientists with a good command of the English language to refrain from stylistic turns and instead focus on clarity. This man had clearly struggled many times with reading in English and I suspect the Goldilocks reference would be one of those times that when he finally was able to piece together what it meant, he would be frustrated beyond belief that it was not given a label that was direct and clear to non-native English speakers.

Thank you for this important comment. We do want our paper to be accessible to a broad audience, including non-native English speakers. We therefore changed “goldilocks” to “optimal” throughout.

3. Also related to the goldilocks zone, I think that figures 3D and 3F could be reorganized or in some way combined to emphasize that the goldilocks zone is defined by effects in both the intact and double cut nerve data.

Done. We added an arrow and box to make this more obvious to the reader.

4. This sentence was interesting to me and I’d like a little more explanation and/or speculation: “When the animal was grounded, substantial dilation occurred during VNS with intact, single or double cut nerve (Fig. S3G-L), as well as substantial offset dilation subsequent to the end of each VNS train (Fig. S3G-L), which was not observed in ungrounded conditions (Fig. S3A-F).”

We have analyzed this more extensively, and now find that phase-locked eye movements are a better readout of off-target effects due to current spread from the animal being grounded (Fig. S4). We have extensively revised, accordingly, and this should be much clearer.

5. I'm embarrassed to ask this, but I'm never used mediation analysis before, and am only moderately familiar with how it works. How is the directionality determined? That is, how does the model of VNS driving ACh driving pupil dilation account for more variance than a model in which VNS drives pupil dilation which drives ACh?

Mediation analysis is purely correlational, so there is unfortunately no information about directionality. We here used the mediation analysis not to arbitrate between directionalities, but to address whether pupil can be used a reliable readout of VNS-evoked ACh responses. A correlational framework is well suited for this, and note that the causative manipulation of the vagus nerve (stimulation) firmly places the vagus stim parameters as the independent variable in the mediation analysis. Put another way, our analysis is agnostic about what the true circuit is linking the vagus nerve, pupil, and cortical acetylcholine. For example, the vagus could drive locus coeruleus, which in turn drives pupil dilation and basal forebrain. Or another neuromodulator (e.g. serotonin) or more distributed circuit could be the intermediary. You mentioned Granger causality, but to our understanding it only works with ongoing time-series. And we would be wary of any analysis that would seem to suggest causality or directionality, because we firmly believe that much more extensive characterization of the circuit linking the vagus nerve to pupil and basal forebrain (and causative manipulations of that circuit) are really what's required to get at the directionality you are interested in. Such experiments would be far beyond the scope of this paper.

References

- Cazettes, F., Reato, D., Morais, J. P., Renart, A., & Mainen, Z. F. (2020). Phasic activation of dorsal raphe serotonergic neurons increases pupil-linked arousal. *BioRxiv*, 2020.06.25.171637. <https://doi.org/10.1101/2020.06.25.171637>
- de Gee, J. W., Colizoli, O., Kloosterman, N. A., Knapen, T., Nieuwenhuis, S., & Donner, T. H. (2017). Dynamic modulation of decision biases by brainstem arousal systems. *ELife*, 6, 309.
- Joshi, S., Li, Y., Kalwani, R. M., & Gold, J. I. (2016). Relationships between Pupil Diameter and Neuronal Activity in the Locus Coeruleus, Colliculi, and Cingulate Cortex. *Neuron*, 89(1), 221–234.
- Reimer, J., McGinley, M. J., Liu, Y., Rodenkirch, C., Wang, Q., McCormick, D. A., & Tolia, A. S. (2016). Pupil fluctuations track rapid changes in adrenergic and cholinergic activity in cortex. *Nature Communications*, 7, 13289.

Reviewer #1 (Remarks to the Author):

The Authors has performed a thorough revision and answered most of my concerns. At the same time, I am not sure why the plot corresponding to the mediation analysis has changed, and what are the implications of this – see below.

Major point

I am confused about the mediation analysis now. I thought c was VNS on pupil, in a way broken down to indirect ($a \times b$) and direct (c') – suggested by the picture in previous Fig.7D, now Fig.6E and supported by the equations in Methods; thus the point being $a \times b > c'$ (pink over grey). I am confused why this analysis has changed, and wonder whether the conclusion, spelled out the same way as in the previous version, has not changed as well. If so, I must be reading this figure wrong. Please clarify this.

Minor points

1. Line 43 – ‘However, the course occlusion’ – typo?
2. Line 177 – ‘than the on double cut nerve’ – revise
3. To me, the locomotion analysis (from line 217) does not really respond to the subsection title: ‘VNS effects on pupil-indexed neuromodulation do not depend on locomotion’. Actually, the analysis corresponding to that title seems to be in the beginning of the next subsection. Changing the title or reorganizing the text could help.
4. Figure 7 still says ‘goldilocks’. It works for me, but probably better to be consistent.

Reviewer #2 (Remarks to the Author):

The authors have sufficiently addressed all my previous points.

We thank the reviewers for their thoughtful comments and continued enthusiasm for our work. Please see below our detailed responses (in black) to the reviewer comments (in blue). We hope that you will now find our manuscript suitable for publication in *Nature Communications*.

Sincerely, on behalf of all authors,
Jan Willem de Gee, Zakir Mridha, and Matthew McGinley

Reviewer #1 (Remarks to the Author):

The Authors has performed a thorough revision and answered most of my concerns. At the same time, I am not sure why the plot corresponding to the mediation analysis has changed, and what are the implications of this – see below.

Thank you for critically reevaluating our manuscript. We are pleased you found the revision to be thorough and that we answered most of your concerns.

Major point

I am confused about the mediation analysis now. I thought c was VNS on pupil, in a way broken down to indirect ($a \times b$) and direct (c') – suggested by the picture in previous Fig.7D, now Fig.6E and supported by the equations in Methods; thus the point being $a \times b > c'$ (pink over grey). I am confused why this analysis has changed, and wonder whether the conclusion, spelled out the same way as in the previous version, has not changed as well. If so, I must be reading this figure wrong. Please clarify this.

The reviewer correctly interpreted the mediation analysis: the total effect (c -path) is broken down into indirect ($a \times b$ path) and direct (c' -path) effects. Indeed, the mediation analysis result changed slightly from the first draft as a result of our improved pupil-fitting neural network. This resulted in a stronger relationship between VNS and evoked pupil responses (the c -path), and a slightly smaller mediation effect ($a \times b$ path). In order to more accurately describe the effect, we now changed:

“We found that the indirect path of VNS driving ACh, in turn driving pupil dilation, statistically mediates the majority of the apparent effect of VNS on pupil.”

To the following:

“We found that the indirect path of VNS driving ACh, in turn driving pupil dilation, partially mediates the apparent effect of VNS on pupil.”

In more detail, the difference is due to us having improved the pupil-fitting neural network and the pupil preprocessing pipeline: we now more accurately detect blinks and interpolate across the missing data. This resulted in a stronger relationship between VNS and evoked pupil responses (c -path): explained variance is now 37.81% (Figure 6B), compared to 32.29% in our first draft (previous Figure 7B). Please note that the b -path and c' -path coefficients are simultaneously obtained through the multiple regression of evoked pupil responses on evoked calcium responses and VNS (eq. 6). In other words, VNS and evoked calcium responses *compete* to explain variance in the evoked pupil responses. The result of VNS now being able to do this more effectively with the improved network weakens the b -path, and therefore weakens the indirect path ($a \times b$).

This quantitative change does not change our overall conclusion. For example, in the original as well as the current draft, the c'-path was significant, and thus the mediation was “partial” in both cases. In the current draft, still a *substantial* part of the relationship between VNS and evoked pupil responses is mediated by the recruitment of the cholinergic system (the indirect path is 49,7% of the total effect). We think that ~50% of the total effect, in combination with statistical significance, for non-invasively explaining activity in a small number of randomly selected axons in cortex is a strong result, and that our overall conclusion stands: “We conclude that pupil dilation provides a sensitive readout of VNS-evoked cholinergic modulation of cortex.”

Minor points

1. Line 43 – ‘However, the course occlusion’ – typo?

Changed to: ‘However, the coarse occlusion [...].’

2. Line 177 – ‘than the on double cut nerve’ – revise

Changed to ‘[...] than on double cut nerve [...].’

3. To me, the locomotion analysis (from line 217) does not really respond to the subsection title: ‘VNS effects on pupil-indexed neuromodulation do not depend on locomotion’. Actually, the analysis corresponding to that title seems to be in the beginning of the next subsection. Changing the title or reorganizing the text could help.

Agreed. We changed the title of this section to: “*Graded dependence of locomotion on VNS parameters*”.

4. Figure 7 still says ‘goldilocks’. It works for me, but probably better to be consistent.

This is now changed.

Reviewer #2 (Remarks to the Author):

The authors have sufficiently addressed all my previous points.

We are glad to hear that we have sufficiently addressed your points.

Reviewer #1 (Remarks to the Author):

Thank you for explaining me why the mediation analysis results were somewhat different across the versions. I think the modified description accurately reflects the result and I have no further concerns. I congratulate the Authors on the thorough revision and suggest publication.